# Atomic-precision Pt$_6$ nanoclusters for enhanced hydrogen electro-oxidation

Xiaoning Wang[1,5], Lianming Zhao[1,5], Xuejin Li[1,5], Yong Liu[2], Yesheng Wang[1], Qiaofeng Yao[3,4], Jianping Xie [3,4], Qingzhong Xue[1], Zifeng Yan[1], Xun Yuan [2✉] & Wei Xing [1✉]

The discord between the insufficient abundance and the excellent electrocatalytic activity of Pt urgently requires its atomic-level engineering for minimal Pt dosage yet maximized electrocatalytic performance. Here we report the design of ultrasmall triphenylphosphine-stabilized Pt$_6$ nanoclusters for electrocatalytic hydrogen oxidation reaction in alkaline solution. Benefiting from the self-optimized ligand effect and atomic-precision structure, the nanocluster electrocatalyst demonstrates a high mass activity, a high stability, and outperforms both Pt single atoms and Pt nanoparticle analogues, uncovering an unexpected size optimization principle for designing Pt electrocatalysts. Moreover, the nanocluster electrocatalyst delivers a high CO-tolerant ability that conventional Pt/C catalyst lacks. Theoretical calculations confirm that the enhanced electrocatalytic performance is attributable to the bifold effects of the triphenylphosphine ligand, which can not only tune the formation of atomically precise platinum nanoclusters, but also shift the $d$-band center of Pt atoms for favorable adsorption kinetics of *H, *OH, and CO.

[1] State Key Laboratory of Heavy Oil Processing, China University of Petroleum, Qingdao 266580, P. R. China. [2] School of Materials Science and Engineering, Qingdao University of Science and Technology, Qingdao 266042, P. R. China. [3] Department of Chemical and Biomolecular Engineering, National University of Singapore, 4 Engineering Drive 4, Singapore 117585, Singapore. [4] Joint School of National University of Singapore and Tianjin University, International Campus of Tianjin University, Binhai New City, Fuzhou 350207, P. R. China. [5]These authors contributed equally: Xiaoning Wang, Lianming Zhao, Xuejin Li. ✉email: yuanxun@qust.edu.cn; xingwei@upc.edu.cn

Alkaline exchange membrane fuel cells (AEMFCs) have been nominated as one of the most promising energy suppliers due to their green attributes and high energy efficiency[1,2]. Although AEMFCs are able to deliver Pt-like performance in economic electrocatalysis of cathodic oxygen reduction reaction (ORR), the same cannot be said for the anodic hydrogen oxidation reaction (HOR)[3–12]. Indeed, the sluggish HOR kinetics in alkaline media (in comparison to that in acidic media) demands a much higher Pt loading (almost ten times higher than that in acids) and thus inevitably results in high costs[13–17]. Despite intensive investigations in cost-effective alternative electrocatalysts in the past several decades, Pt remains the best HOR electrocatalyst, inherently owing to its near-optimal hydrogen binding energy (HBE). However, the conventional Pt/C electrocatalyst is suffering from high cost, poor anti-poisoning capability, and insufficient stability, which bring about additional obstacles for industrial applications of Pt-based catalysts in alkaline HOR[18]. Therefore, it is highly desirable to engineer the Pt catalysts at the atomic level for higher atomic utilization, enhanced anti-poisoning capability, and better stability.

Atomically precise metal nanoclusters (NCs), which bridge metal single atoms (SAs) and large-sized metal nanoparticles (NPs, >2 nm), have emerged as functional materials for addressing various challenges in diverse fields, due to their unique molecule-like structural properties (e.g., ultrasmall size (<2 nm), ultrahigh surface-to-volume ratio, atomic-precision structure, and rich surface chemistry) and physicochemical properties (e.g., luminescence, chirality, and discrete HOMO-LUMO transition)[19–29]. For example, ultrasmall metal NCs (e.g., Au, Pt, Pd, and their alloy analogues) have been employed as model catalysts for $CO_2$ reduction reaction[30], alcohols oxidation[31], hydrogen evolution reaction (HER)[32], and ORR[33,34], which makes the understanding on the catalytic mechanism as well as the design of nanocatalysts stepping from nanoscale into the molecular level[35]. Specifically, a precise metal cluster containing only a few atoms could have a discrete energy band structure, which would impel the $d$-band center to shift with the number of metal atom for favorable adsorption kinetics modulation of intermediates with purpose[36]. From this viewpoint, atomically precise metal NCs may provide a good opportunity for us to address the aforementioned challenges in the design of Pt catalysts, especially considering NC's powerful capabilities in manipulating atom-precision structure and regulating the interfacial catalytic behavior.

However, the synthesis of Pt NCs with definite Pt atom number and decent surface chemistry is a long-standing challenge in the cluster research community. This is most probably due to the vibrant catalytic activity of freshly generated Pt atoms that induce in-situ deposition and catalytic decomposition of reducing agents, like $NaBH_4$ in most reduction-growth strategies, giving rise to wild reduction kinetics that makes the controllable growth of Pt NCs unsuccessful. In addition, the surface chemistry of Pt NCs should also be engineered for the design of Pt NCs electrocatalysts, because the organic ligands not only influence the stability as well as the catalytic activity of Pt NCs through regulating the atomic packing structure and dictating the diffusion of reactants on the surface of Pt NCs, but also impact the adsorption/desorption competition between reactants and poisoning species on the catalytic active sites of Pt NCs[37].

Herein, we report the design of $Pt_6(PPh_3)_4Cl_5$ NCs toward the alkaline HOR catalysis by utilizing a mild reducing agent, borane-tert-butylamine (TBAB for short), in a less polar medium, and choosing triphenylphosphine ($PPh_3$) molecules as a benign protecting agent. Benefiting from the self-optimized ligand effect and precise atomic structure, the catalyst exhibits excellent catalytic activity and high stability and outperforms Pt SAs and Pt NPs analogues. Moreover, it also exhibits high tolerance against CO impurity by weakening the binding energy of CO intermediates. Theoretical simulations highlight the structural significance of the atomic-precision $Pt_6$ NCs and further uncover the distinctive catalytic mechanism of molecular-structure $Pt_6$ NCs for enhancing the whole HOR performance.

## Results and discussion

**Synthesis and characterization of catalyst**. Regarding the NCs synthesis, it is well-documented that a mild reduction environment is crucial to the precise size control of metal NCs[38,39], which may be even more pivotal in the case of Pt NC synthesis. This is due to the superior catalytic performance of freshly formed Pt, which could accelerate the decomposition of $NaBH_4$, yielding too fast reduction kinetics. Once introducing the strong reducing agent $NaBH_4$ into the solution containing Pt precursor (Supplementary Fig. 1 and Supplementary Movie 1), abundant bubbles of $H_2$, which is the product of $NaBH_4$ decomposition and acts as active species in the subsequent reduction reaction, could be observed. A similar phenomenon was not observed during the synthesis of Au and Ag NCs within commonly used $NaBH_4$-reduction systems[19,38], evidencing the self-catalytic feature of Pt in the $NaBH_4$ reduction system. Owing to the wild and thus less controllable reduction kinetics inherently induced by the self-catalytic habit of Pt, no molecularly pure Pt NCs but polydisperse Pt particles (40 to 90 nm) could be obtained by typical $NaBH_4$ reduction methods (Supplementary Fig. 2).

In order to achieve atomic precision in Pt NC synthesis, as shown in Fig. 1a, we employed a mild reducing agent TBAB to replace the $NaBH_4$ to minimize the catalytic effect of Pt species (Route I), thus slowing down the reduction kinetics for Pt NC growth. The reduction kinetics was further retarded by using less polar solvent (e.g., toluene and ethanol) to replace polar water, thereby creating a mild reduction environment[39]. Besides, $PPh_3$ ligands with the proper affinity to Pt are selected for the formation of atomically precise Pt NCs. Here, $PPh_3$ can serve as σ donors and π acceptors, leading to the formation of $p\pi–d\pi$ P–Pt bonds with moderate binding energy to avert poisoning of Pt active sites[40,41]. Such a three-pronged approach results in $PPh_3$-protected Pt NCs with atomic precision. By simply tuning the experimental parameters, the $PPh_3$-protected Pt SAs and large-sized Pt NPs are also obtained as references for the latter electrocatalytic test.

Upon acquiring Pt NCs, ultraviolet-visible (UV-vis) absorption spectroscopy, as well as electrospray ionization mass spectrometry (ESI-MS) were firstly performed to analyze their size. As shown in Fig. 1b, two absorption bands at 335 and 385 nm are observed in the UV-vis spectrum of Pt NCs, which implies the formation of ultrasmall Pt NCs with molecule-like optical absorption features[39]. The ESI-MS result of Pt NCs shows an intense peak centered at $m/z = 2395$ (Fig. 1c), which can be assigned to $Pt_6(PPh_3)_4Cl_5$ carrying +1 charge. The experimentally obtained isotope pattern of $Pt_6(PPh_3)_4Cl_5$ is in good agreement with the simulated one (inset in Fig. 1c), corroborating the formation of $Pt_6(PPh_3)_4Cl_5$ NCs (denoted as $Pt_6$NCs)[42]. Such a result also highlights the importance of precisely controlling the reduction kinetics in dictating the size monodispersity of atomically precise Pt NCs. By contrast, the sample prepared without introducing the reducing agent, TBAB, presents a distinctive peak at 390 nm in the UV-vis spectrum (Supplementary Fig. 3), and the intense peak centered at $m/z = 754$ in the ESI-MS spectrum (Supplementary Fig. 4) corresponding to $Pt_1(PPh_3)_2Cl$, evidencing the formation of single-atom Pt catalyst (denoted as $Pt_1$SAs). In addition, the as-fabricated $Pt_6$NCs display

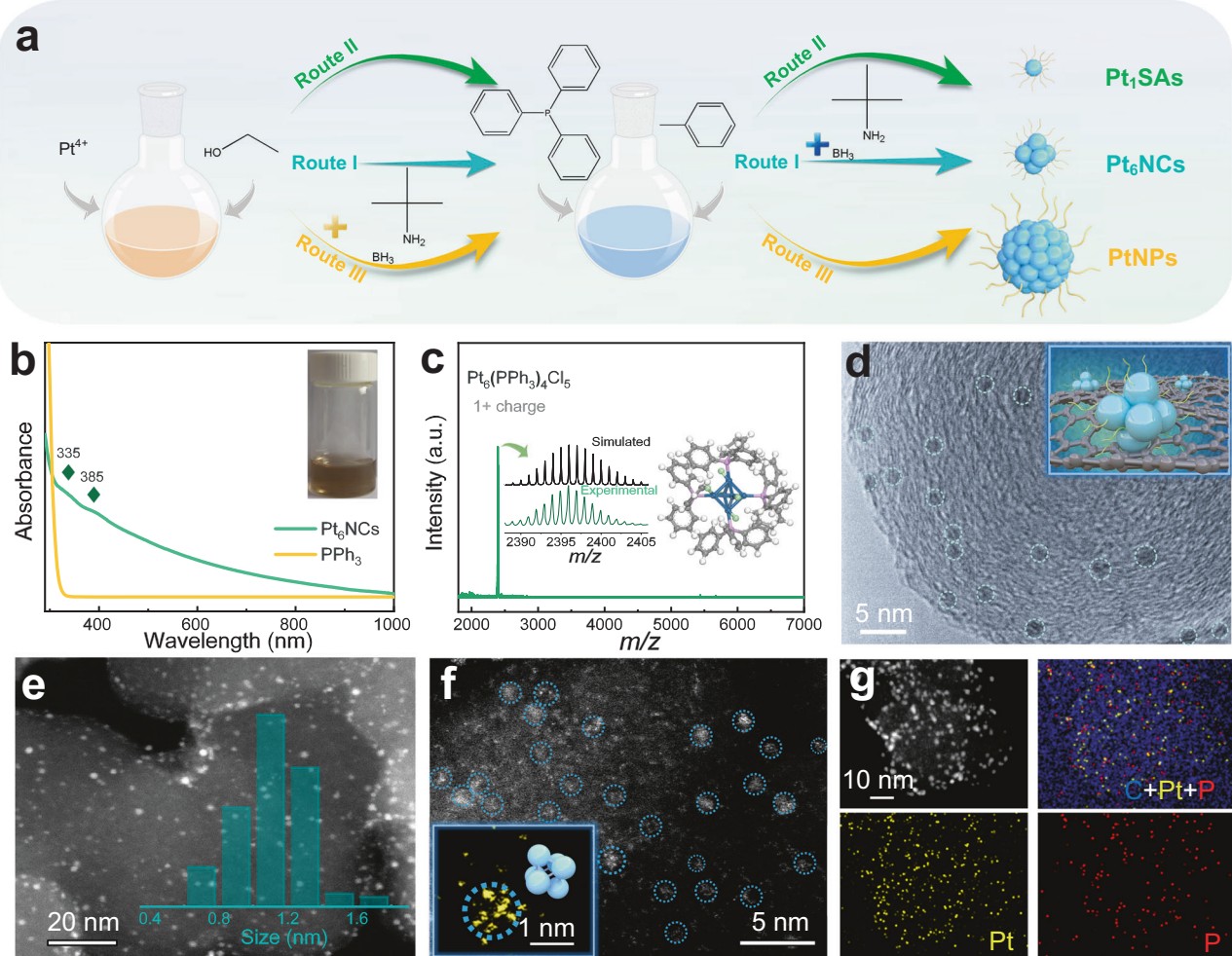

**Fig. 1 Synthesis, morphological, and structural characterization of Pt₆NCs. a** Schematic representation of the synthetic routes for PPh₃-protected Pt-based catalysts: Pt₁SAs, Pt₆NCs, and large-sized PtNPs. **b** UV-vis absorption spectra of Pt₆NCs and PPh₃; inset: the photographic image of the Pt₆NCs suspension. **c** ESI-MS spectrum of Pt₆NCs (measured in positive mode); inset: (left) experimentally obtained (green curve) and simulated (black curve) isotope patterns of Pt₆(PPh₃)₄Cl₅; (right) a simulated structure model of a Pt₆NC; The gray, white, blue, pink, and reseda spheres represent C, H, Pt, P, and Cl atoms, respectively. **d** TEM image of Pt₆NCs/C; inset: schematic diagram of carbon-supported Pt₆NCs. **e** HAADF-STEM image of Pt₆NCs/C; inset: size distribution of Pt₆NCs. **f** Aberration-corrected HAADF-STEM image of Pt₆NCs/C; inset: the magnified image of a Pt₆NCs and corresponding model. **g** Corresponding element maps of Pt₆NCs/C showing distributions of C (blue), Pt (yellow), and P (red), respectively.

good stability, and there is no obvious change in the ESI-MS spectra for the Pt₆NCs after three weeks of storage at 4 °C without N₂ protection (Supplementary Fig. 5).

Prior to the electrochemical test, the Pt₆NCs were firstly deposited on commercially available carbon black by impregnation adsorption, and the as-formed Pt₆NCs/C was characterized by transmission electron microscopy (TEM). As shown in Fig. 1d, individual NC is visualized as a separated dark dot with a typical diameter of ~1 nm (inset in Fig. 1e). The formation of ~1 nm NCs was also supported by high-angle annular dark-field scanning transmission electron microscopy (HAADF-STEM), where NCs appear as bright dots (Fig. 1e). Aberration-corrected HAADF-STEM images (Fig. 1f and inset) confirm the atom-precise synthesis of clusters based on the expected structural models. The above-mentioned morphological characterizations demonstrate that Pt₆NCs are well dispersed (without notable aggregation or agglomeration), which should be attributed to the steric effects of PPh₃ ligands and the immobilization effects of the carbon support. Elemental maps of the Pt₆NCs/C (Fig. 1g) confirm the uniform distribution of Pt and P elements across the surface of the carbon support. The Pt loading of Pt₆NCs/C was measured to

be 2.5 wt.% by thermogravimetric analysis (TGA; Supplementary Fig. 6a). In comparison, TEM and HAADF-STEM images of carbon-supported Pt₁SAs (denoted as Pt₁SAs/C) reveal the formation of atomically dispersed Pt₁ atoms with only a few sporadic NCs or NPs (Supplementary Fig. 7). TEM image of carbon-supported PtNPs (denoted as PtNPs/C) shows that the average size of NPs reaches up to 4.4 nm (Supplementary Fig. 8a). The lattice fringe of the Pt NPs (Supplementary Fig. 8b) has a regular spacing of 0.22 nm, corresponding well to the interplanar spacing of (1 1 1) plane of crystalline Pt. The Pt loading of Pt₁SA/C and PtNPs/C were calculated as 2.2 wt.% and 5.4 wt.% by TGA (Supplementary Fig. 6), respectively. Furthermore, we also attempted to synthesize Pt NCs by using thiolate ligands (e.g., glutathione or GSH for short), which gives rise to Pt NCs with a much larger diameter of 1.9 nm (Supplementary Fig. 9), implying that the interaction between ligands and Pt ions is crucial in determining the size and monodispersity of Pt NCs.

Powder X-ray diffraction (PXRD) analysis (Fig. 2a) was carried out to confirm the crystalline structure of Pt-based catalysts. No characteristic peak associated with Pt species can be detected in the spectra of Pt₆NCs/C and Pt₁SA/C, thereby ruling out the

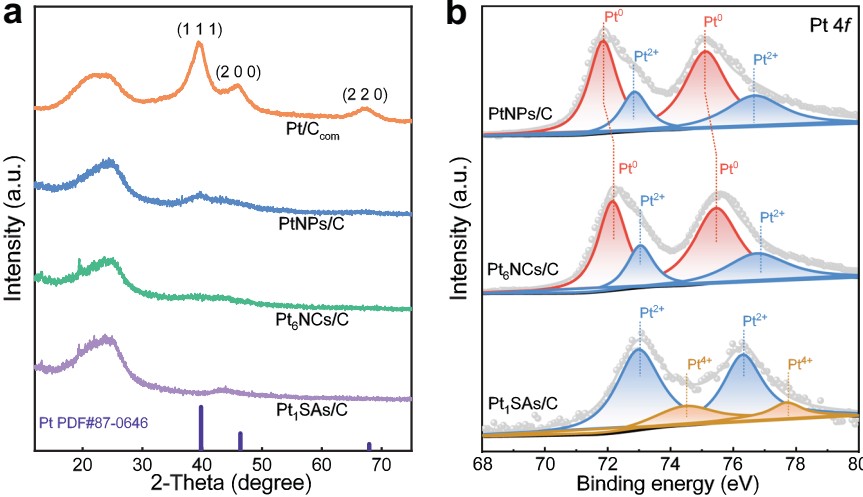

**Fig. 2 Structural characterizations of carbon-supported Pt-based catalysts. a** XRD patterns of $Pt_1SAs/C$, $Pt_6NCs/C$, PtNPs/C, and $Pt/C_{com}$. **b** Pt 4$f$ XPS spectra comparison of $Pt_1SAs/C$, $Pt_6NCs/C$, and PtNPs/C.

formation of crystalline Pt NPs. Three diffraction peaks corresponding to Pt (1 1 1), (2 0 0), and (2 2 0) planes can be observed in the spectrum of PtNPs/C, which are similar to those observed in commercial Pt/C (20 wt.%, denoted as $Pt/C_{com}$), indicating the formation of Pt nanocrystals with long-range order. X-ray photoelectron spectroscopy (XPS) analysis was further performed to examine the electronic properties and chemical states of as-prepared Pt catalysts. The Pt 4$f$ XPS spectrum of $Pt_6NCs/C$ manifests Pt $4f_{7/2}$ and Pt $4f_{5/2}$ at 72.2 eV and 75.4 eV (Fig. 2b), respectively. The Pt $4f_{7/2}$ and Pt $4f_{5/2}$ peaks can be deconvoluted into two spin-orbit doublets, indicating the dominant population of $Pt^0$ and the minor population of $Pt^{2+}$ (Fig. 2b). Pt in PtNPs/C exhibits similarly combined valences of $Pt^0$ and $Pt^{2+}$, while Pt in $Pt_1SAs/C$ existed mainly $Pt^{2+}$ and $Pt^{4+}$. Notably, the binding energy of Pt 4f in $Pt_6NCs/C$ presents a remarkable positive core-level shift of 0.4 eV as compared with that in PtNPs/C, inferring that more electron transfer from Pt atoms to P atoms due to the stronger orbital interactions[43]. This assertion is supported by our control experiments, where ligand removal by traditional calcination treatment can lower the binding energy of Pt 4f in $Pt_6NCs/C$ catalysts (Supplementary Fig. 10a). This readily means that the PPh₃ ligands have strong interactions with Pt centers and thus can significantly alter the electron distribution of the latter.

**Evaluation of HOR performance**. The HOR activity of Pt-based catalysts was evaluated in $H_2$-saturated 0.1 M KOH aqueous solution. $Pt/C_{com}$ (an optimal loading of 10 µg cm$^{-2}$, on the Pt basis thereafter) was introduced as the benchmark for comparison (Supplementary Fig. 11a). Generally, more Pt usage induces higher catalytic activity before reaching the optimum loading amount, and here it is demonstrated that further increasing the catalyst loading of $Pt_6NCs/C$ over 5 µg cm$^{-2}$ would not lead to additional performance gain (Supplementary Fig. 11b). Therefore, the mass loading of $Pt_6NCs/C$ was set to be 5 µg cm$^{-2}$ to maximize the catalytic performance with the minimized Pt loading. As shown in Fig. 3a, the polarization curve of $Pt_6NCs/C$ quickly rises to the saturation current density of 2.7 mA cm$^{-2}$ at an overpotential of 50 mV (vs. Reversible hydrogen electrode (RHE)). By contrast, the anodic current density of $Pt/C_{com}$ increases more mildly (1.8 mA cm$^{-2}$ at an overpotential of 50 mV (vs. RHE)). Meanwhile, $Pt_6NCs/C$ displays a much higher anodic current than $Pt_1SAs/C$ and PtNPs/C at the kinetic control regions, implying that the atomically precise six-Pt-atom

structure promotes their electrocatalytic activity on HOR. In contrast to the inconspicuous anodic current across the whole potential range in the Ar-saturated 0.1 M KOH (Supplementary Fig. 12), $Pt_6NCs/C$ shows a fast anodic current response in the presence of $H_2$ above 0 V (vs. RHE), suggesting that the anodic current is derived from the oxidation of $H_2$.

The mass-specific activity of $Pt_6NCs/C$ was derived to quantitatively analyze the HOR activity of the catalysts. Figure 3b displays the HOR polarization curves under different electrode rotation rates. By plotting and fitting the inverse of the current density ($j^{-1}$) at an overpotential of 50 mV (vs. RHE) versus the inverse of the square root rotation rate ($\omega^{-1/2}$), a straight line with a slope of 5.08 cm$^2$ mA$^{-1}$ s$^{-1/2}$ is obtained, agreeing well with the Koutecky–Levich equation (see details in the "Electrochemical measurements"). The as-calculated slope is approaching to the theoretical value of 4.87 cm$^2$ mA$^{-1}$ s$^{-1/2}$ for the two-electron HOR process[44,45]. By extrapolating the $\omega^{-1/2}$ to 0, the kinetic current density ($j_k$) of 18.2 mA cm$^{-2}$ and the mass-specific kinetic current ($j_{k,m}$) of 3658 A g$^{-1}$ are obtained. Remarkably, the mass-specific activity of $Pt_6NCs/C$ is almost 9.1 times higher than that of $Pt/C_{com}$ (402 A g$^{-1}$) and 7.4 times higher than that of PtNPs/C (496 A g$^{-1}$), as shown in Supplementary Fig. 13 and Supplementary Table 1. As-synthesized $Pt_6NCs/C$ catalysts also exhibit superior activity than most of the previously reported platinum group metal (PGM) catalysts (Supplementary Table 1). To the best of our knowledge, this is the highest attainable value among all monometallic HOR electrocatalysts under the tested conditions.

Tafel plots of Pt-based catalysts originated from the $j_k$ were depicted according to the Bulter-Volmer equation in Fig. 3c. The $j_k$ at an overpotential of 20 mV (vs. RHE) is 3.62 mA cm$_{disk}^{-2}$ for $Pt_6NCs/C$, which is much higher compared to those of $Pt_1SAs/C$, PtNPs/C, and $Pt/C_{com}$ catalysts. Since the HER and HOR branches are symmetrical with each other, the HOR reaction mechanisms of all catalysts are determined to be a Tafel ($H_2 + 2* \leftrightarrows 2*H$, * represents adsorption site) – Volmer ($*H + OH^- \leftrightarrows H_2O + e^- + *$) process and the Volmer step is the rate-determining step. The exchange current density ($j_0$) is also determined from the linear fitting of the micro-polarization region (Fig. 3d). The mass-normalized $j_0$ ($j_{0,m}$) of $Pt_6NCs/C$ (646.3 A g$^{-1}$) is about 29.4, 3.9, and 5.7 times higher than those of $Pt_1SAs/C$ (22.0 A g$^{-1}$), PtNPs/C (164.5 A g$^{-1}$), and $Pt/C_{com}$ (113.1 A g$^{-1}$), respectively. The area-specific exchange current density ($j_{0,s}$) values are obtained by normalization of the $j_{0,m}$ according to the corresponding

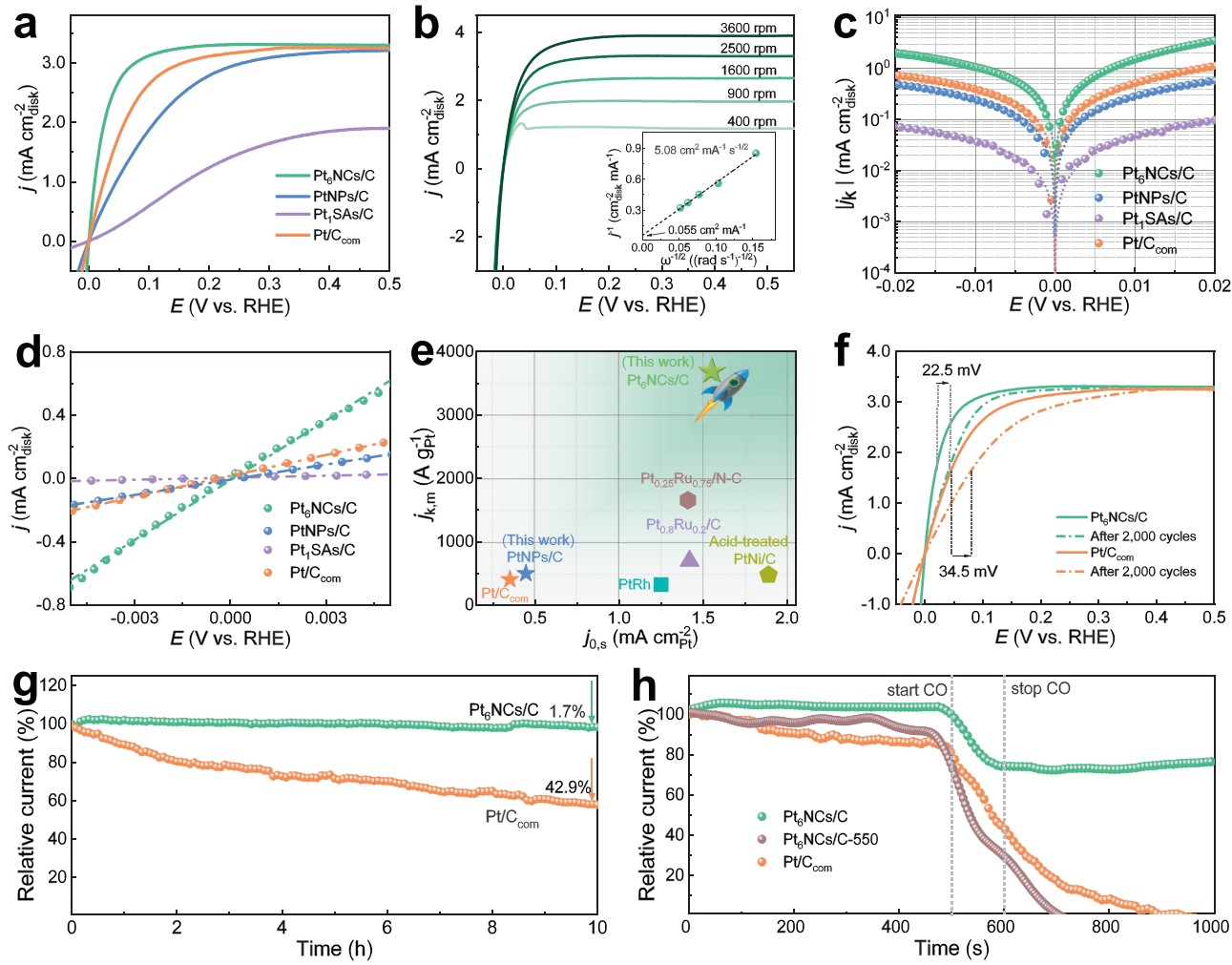

**Fig. 3 HOR performance catalyzed by Pt-based catalysts. a** HOR polarization curves of $Pt_1SAs/C$, $Pt_6NCs/C$, $PtNPs/C$, and $Pt/C_{com}$ catalysts in $H_2$-saturated aqueous solution of 0.1 M KOH with a rotating rate of 2500 rpm at a scan rate of 5 mV s$^{-1}$. **b** HOR polarization curves of $Pt_6NCs/C$ catalyst at different rotation rates. Inset: the Koutecky–Levich plot of $Pt_6NCs/C$ catalyst at an overpotential of 50 mV (vs. RHE). **c** Representative HOR/HER Tafel plots of kinetic current density ($j_k$) for $Pt_1SAs/C$, $Pt_6NCs/C$, $PtNPs/C$, and $Pt/C_{com}$ catalysts. Symbols are calculated $j_k$ values, and lines are fitted results based on the Volmer-Butler equation. **d** Linear current potential region around the equilibrium potential of HOR/HER of $Pt_1SAs/C$, $Pt_6NCs/C$, $PtNPs/C$, and $Pt/C_{com}$ catalysts conducted at a scan rate of 5 mV s$^{-1}$ with the rotation rate at 2500 rpm. The dotted lines indicate the linear fitting of the data. **e** Comparison of the mass-specific kinetic current ($j_{k,m}$) at an overpotential of 50 mV (vs. RHE) and the area-specific exchange current ($j_{0,s}$) with those of previous reports. **f** HOR polarization curves for $Pt_6NCs/C$ and $Pt/C_{com}$ before (solid lines) and after (dashed lines) accelerated durability test, respectively. **g** Relative current-time chronoamperometry response of $Pt_6NCs/C$ and $Pt/C_{com}$ catalysts in an $H_2$-saturated aqueous solution of 0.1 M KOH at an overpotential of 60 mV (vs. RHE). **h** CO poisoning experiments of $Pt_6NCs/C$, $Pt_6NCs/C$-550, and $Pt/C_{com}$ catalysts at an overpotential of 100 mV (vs. RHE).

electrochemically active surface area (ECSA) value (Supplementary Table 1 and Supplementary Fig. 14). $Pt_6NCs/C$ catalyst exhibits a much higher $j_{0,s}$ value (1.546 mA cm$^{-2}$) compared to those of $Pt_1SAs/C$ (0.149 mA cm$^{-2}$), $PtNPs/C$ (0.446 mA cm$^{-2}$), and $Pt/C_{com}$ (0.351 mA cm$^{-2}$) catalysts (Fig. 3e), implying that $Pt_6NCs/C$ possesses better catalytic activity toward HOR than $Pt_1SAs/C$ and $PtNPs/C$. These results unambiguously indicate $Pt_6NCs$ as the optimal size for electrocatalytic HOR, in comparison to their smaller (SAs) and larger (NPs) counterparts.

To shed more light on the effect of $PPh_3$ ligands on HOR catalytic activity, we performed a thermal annealing treatment on $Pt_6NCs/C$ at 550 °C under Ar atmosphere to remove the ligands (denoted as $Pt_6NCs/C$-550). It is found that $Pt_6NCs/C$-550 can largely retain the discrete and tiny morphology of Pt NCs, with only negligible agglomeration observed (Supplementary Fig. 15). The calcination process eliminated $PPh_3$ ligands (Supplementary Fig. 10b), excluding any possible impact from ligands on the

catalytic activity of NCs. $Pt_6NCs/C$-550 possesses inferior HOR activity ($j_{k,m}$ = 2128 A g$^{-1}$ and $j_{0,m}$ = 462.2 A g$^{-1}$) than $Pt_6NCs/C$ (Supplementary Fig. 16), which may be attributed to the slight aggregation of Pt NCs and the removal of the $PPh_3$ ligands. These results indicate that instead of blocking the active sites of Pt nanomaterials as previously reported, the $PPh_3$ ligands anchored on the surface of $Pt_6$ NCs can not only maintain the molecular structure of the Pt NCs, but also improve their catalytic activity probably by regulating the molecule-like catalytic behavior (e.g., adsorption/desorption kinetics of both reactants and the as-produced reaction intermediates, and reactivity of active sites). We termed the phenomenon above as the self-optimizing ligand effect, which is derived from the interaction between inorganic Pt and organophosphorus ligand structures, including steric repulsion and electronic effects[37]. In contrast to the $PPh_3$ ligand, the hydrosoluble GSH ligand containing the toxicant S element presents an adverse effect on the catalytic activity of Pt NCs

(Supplementary Fig. 17), indicating that the ligand attributes play a crucial role in catalyzing HOR.

Catalytic durability is another important attribute dictating the usefulness of Pt NCs in HOR. Firstly, an accelerated durability test (ADT) was performed on $Pt_6NCs/C$ catalyst by 2000 repetitive cyclic voltammetry (CV) scans in $H_2$-saturated 0.1 M KOH aqueous solution (Supplementary Fig. 18a). It is found that the ECSA of $Pt_6NCs/C$ decreases by 3.6% and 15.2% after 1000 and 2000 cycles (Supplementary Fig. 18b). After 2000 cycles, the $Pt_6NCs/C$ manifests a 22.5 mV increase in half-wave potential, versus 34.5 mV observed on $Pt/C_{com}$ catalyst (Fig. 3f), indicating the robustness of $Pt_6NCs/C$ catalyst. Subsequently, the chronoamperometry was carried out to further evaluate the stability of $Pt_6NCs/C$ (Fig. 3g). After a continuous operation at an overpotential of 60 mV (vs. RHE) for 10 h, the $Pt_6NCs/C$ catalyst showed an inconspicuous degradation and retained 98.3% of the activity (current retention rate). By sharp contrast, a serious deterioration of more than 42% was observed in the catalytic activity of $Pt/C_{com}$ catalyst under the same conditions, indicating the superior durability of $Pt_6NCs/C$ to $Pt/C_{com}$. The surface chemistry and size change of $Pt_6NCs/C$ after the HOR test were also characterized (Supplementary Fig. 19). The similar Pt 4$f$ and P 2$P$ XPS spectra of $Pt_6NCs$ before and after ADT corroborate the good structural stability of $Pt_6NCs$ in the long-term HOR process (Supplementary Fig. 19a, b). HAADF-TEM images reveal that no marked size change could be observed for the $Pt_6NCs/C$ catalyst after ADT (Supplementary Fig. 19c, d). These results unambiguously manifest the superior HOR durability of $Pt_6NCs/C$ catalyst, which originates from the strong interactions of $PPh_3$ ligands with Pt as well as the proper bulkiness of $PPh_3$ ligands, effectively inhibiting the agglomeration of Pt NCs during the electrochemical process.

**CO-tolerance evaluation**. Apart from the activity and long-term durability, CO tolerance is another equally important aspect determining the industrial acceptance of HOR electrocatalysts. Pt is prone to CO poisoning even at low levels of CO (10 ppm) due to the preferential CO binding on Pt, which can block the active sites for hydrogen adsorption and dissociation. The CO tolerance of $Pt_6NCs/C$, $Pt_6NCs/C$-550, and $Pt/C_{com}$ catalysts was evaluated by feeding diluted (5%) CO to the $H_2$-saturated electrolyte for a short period during the chronoamperometric test at an overpotential of 100 mV (vs. RHE)[46]. As depicted in Fig. 3h, $Pt/C_{com}$ catalyst suffered from a rapid decrease in activity and its anodic relative current abruptly dropped from 100% to 0 with the introduction of CO. The current cannot be recovered even the fuel stream is reverted to pure $H_2$, confirming that CO poisoning is irreversible under the tested HOR potential window. By contrast, $Pt_6NCs/C$ catalyst can retain a high HOR activity (current retention rate: 77%) with the same level of CO contamination, evidencing its superior CO-tolerant ability. It should be noted that $Pt_6NCs/C$-550 presents a lower CO tolerance than $Pt/C_{com}$, suggesting the $PPh_3$ ligands play a crucial role in anti-poisoning. Additionally, the position of the CO stripping peak of $Pt_6NCs/C$ exhibits 170 mV negative shifts compared with $Pt/C_{com}$ (Supplementary Fig. 20), which further indicates that $Pt_6NCs/C$ possesses a much lower CO adsorption ability, resulting in the enhanced CO-tolerant ability. We rationalize that the self-optimized ligand effect and the molecular structure of $Pt_6NCs$ offer efficient electron penetration and give rise to sluggish CO adsorption kinetics during the HOR[1].

**Mechanism investigation and computational studies**. The alkaline HOR catalytic mechanism is still under severe debate, but HBE and/or OH binding energy (OHBE) are generally considered

to be the activity descriptor based on theoretical and experimental observations[11]. It is well-documented that the alkaline HOR process follows a Tafel − Volmer or Heyrovsky ($H_2 + OH^- + * \leftrightarrows H_2O + *H + e^-$) − Volmer route, and the Volmer step is regarded as the rate-determining step owing to the strong HBE of PGM. On the basis of the Sabatier principle, weakening HBE in alkaline media could favor higher HOR activity[11,14,18]. One of the salient features of $Pt_6NCs/C$ is that its HBE could be manipulated by both the molecule-like structure and the $PPh_3$ ligands of $Pt_6NCs$. The HBE information of $Pt_6NCs/C$ and $Pt/C_{com}$ could be obtained directly from their CV curves. As shown in Fig. 4a, the CV curves exhibit that the hydrogen desorption peak of $Pt_6NCs/C$ is negatively shifted compared with that of $Pt/C_{com}$ (0.233 vs. 0.293 V (vs. RHE)), implying the smaller HBE as well as the result better HOR activity of $Pt_6NCs/C$ than that of $Pt/C_{com}$.

The molecular structure of the $PPh_3$-protected $Pt_6NCs$ could be largely responsible for the optimization of HBE. Owing to the difficulties in discriminating the molecular structure of $Pt_6NCs$ by ESI-MS analysis, we resorted to density functional theory (DFT) simulations to predict the possible structure of $Pt_6NCs$. Our DFT simulations reveal a stable structure of $Pt_6NCs$ with the lowest energy consisting of partially $P(CH_3)_3-$ and/or Cl-protected six-Pt-atom octahedron, wherein three Pt atoms form Pt–Cl (2.382 Å) and Pt–P (2.236 Å) bonds simultaneously, while the other three Pt atoms form Pt–P (2.223 Å) and Pt–Cl (2.305 Å) bonds individually (Supplementary Fig. 21). The average Pt–Pt bond length amounts to 2.717 Å (Supplementary Fig. 22), which is in good agreement with the reported data in the literature for similar clusters[47]. In addition, the structural models of $Pt_1SAs$ and PtNPs were also simulated by DFT to clarify the correlation of cluster structure and HBE (Supplementary Figs. 23 and 24). However, it should be mentioned that the structures of the $Pt_6NCs$ as well as $Pt_1SAs$ and PtNPs discussed here are simulated based on DFT, and their actual structures still rely on the single-crystal X-ray crystallography, which will be realized with future research efforts. Nevertheless, the DFT simulation could provide useful information to deepen our fundamental understanding of this issue. Of note, based on DFT analysis, the electron gathering area in Pt–P bonds is close to the P atom, while the electron densities around the Pt atom are decreased accordingly (Fig. 4b and c, Supplementary Figs. 25 and 26), indicating that the $PPh_3$ coordination can tune the electronic structure of Pt.

After modulating the electronic configuration, the $d$-band center of Pt sites would show a significant shift. According to the $d$-band theory, the lower $d$-band center far from the Femi level ($E_F$) could be corresponding to the weaker binding strength of intermediate adsorbates, owing to the lower band filling of the antibonding states[48,49]. It is shown that the $d$-band center of the Pt in $Pt_6NCs$ (−2.842 eV) downshifted far away relative to the $E_F$ compared to that of the $Pt_1SAs$ (−0.540 eV) and PtNPs (−2.689 eV) (Fig. 4d), which may be attributed to the intensive $d$-orbital interaction of $Pt_6NCs$, thus revealing the weakened binding strength of *H and CO. In addition, it is found that the Gibbs free energy of *H ($\Delta G_{*H}$ for short) of $Pt_6NCs$ (−0.07 eV; Fig. 4e) on the optimal adsorption site (Pt–P bonding) is very close to the ideal value ($\Delta G_{*H} = 0$) for HOR (Supplementary Fig. 27). By contrast, the $\Delta G_{*H}$ values of $Pt_1SAs$ (0.71 eV) and PtNPs (−0.21 eV) demonstrate larger deviation from the ideal value. In parallel, the formation mechanism and tunable nucleophilicity of $Pt_6NCs$ can be well elucidated by the limited surface negative electrostatic potential obstructed by ligands bonded to Pt atoms (Supplementary Fig. 28)[50], consistent with the DFT analysis. Experimental and simulation results manifest that $Pt_6NCs$ possess the near-optimal HBE due to the self-optimized ligand effect, which can tremendously facilitate the Volmer step for alkaline HOR. Indeed, the attribute of the rich

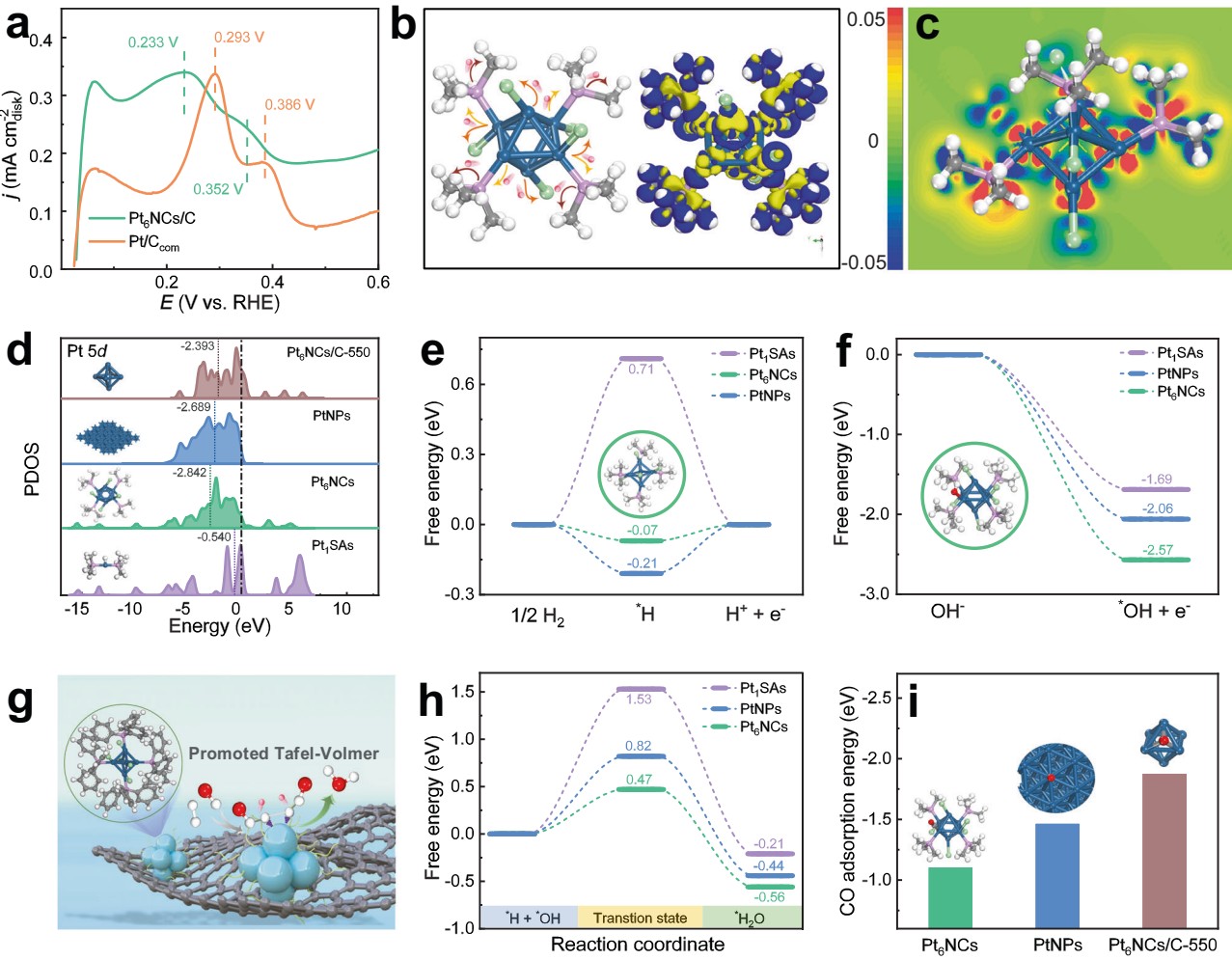

**Fig. 4 Theoretical verification on the structure-activity relationship. a** CV curves of $Pt_6NCs/C$ and $Pt/C_{com}$, showing the hydrogen desorption region. **b** Simulated model of $Pt_6NCs$ (left) and electron density differences of $Pt_6NCs$ (right), where the blue and yellow represent accumulation and depletion of charge, respectively. **c** The slice perspective of differential charge density distribution for $Pt_6NCs$ based on DFT analysis. In the electron density difference maps, the red and blue colors refer to the positive (0.05 e Å$^{-3}$) and negative ($-0.05$ e Å$^{-3}$) values, respectively. **d** The PDOSs of Pt $5d$ in $Pt_1SAs$, $Pt_6NCs$, PtNPs, and $Pt_6NCs/C$-550 (each $d$-band center is marked by a dashed line) with the Fermi level aligned at 0 eV; inset: computed corresponding catalyst models. **e** $\Delta G_{*H}$ on $Pt_1SAs$, $Pt_6NCs$, and PtNPs; inset: computed hydrogen adsorption configuration on $Pt_6NCs$. **f** $\Delta G_{*OH}$ on $Pt_1SAs$, $Pt_6NCs$, and PtNPs; inset: computed hydroxyl adsorption configuration on $Pt_6NCs$. **g** Schematic illustration of HOR catalysis on the $Pt_6NCs/C$. **h** The energy barrier for water formation on $Pt_1SAs$, $Pt_6NCs$, and PtNPs. **i** CO adsorption energy of $Pt_6NCs$, PtNPs, and $Pt_6NCs/C$-550; inset: computed CO adsorption configuration on $Pt_6NCs$, PtNPs, and $Pt_6NCs/C$-550. The gray, red, white, blue, pink, and reseda spheres represent C, O, H, Pt, P, and Cl atoms, respectively.

surface chemistry of metal NCs may offer broad flexibility in regulating the $d$-band center of Pt NCs. For example, DFT simulation reveals that different N-heterocyclic carbene-derivative ligands ranging from strong to weak σ-donors can be employed to tune the $d$-band center of the $Pt_6NCs$, thereby regulating the HBE (Supplementary Fig. 29)[51]. This strategy may provide a feasible research direction in the electronic regulation of catalysts for alkaline HOR catalysis.

Since the adsorbed *OH plays a crucial role to remove the adsorbed *H in the Heyrovsky and Volmer steps, increasing *OH coverage on Pt would greatly benefit alkaline HOR[52,53]. The Gibbs free energy of *OH ($\Delta G_{*OH}$) was analyzed on the *OH adsorption models of $Pt_1SAs$, $Pt_6NCs$, and PtNPs (Supplementary Fig. 30). In alkaline media, the slow HOR rate on the Pt catalyst is closely associated with the inherent weak OH adsorption on Pt (1 1 1) due to weak Pt–OH interaction[54]. However, we found that $Pt_6NCs$ possess a much more negative $\Delta G_{*OH}$ ($-2.57$ eV) than those of PtNPs ($-2.06$ eV) and $Pt_1SAs$ ($-1.69$ eV) (Fig. 4f), suggesting its enhanced adsorptions of *OH to the references.

Furthermore, we conducted CO-stripping tests to reflect the strength of the OHBE on the Pt surface because more *OH can promote the removal of *CO[12]. CO-stripping results (Supplementary Fig. 20) show that $Pt_6NCs/C$ oxidizes *CO at much lower potential compared with $Pt/C_{com}$ ($-170$ mV), implying the stronger OHBE on $Pt_6NCs/C$. Such an enhanced *OH adsorptive behavior would facilitate the capture of *OH species on the surface of $Pt_6NCs$, leading to acceleration of the Volmer step during the alkaline HOR process.

Based on the results above, we reason that the enhanced *OH adsorption and decreased *H adsorption on $Pt_6NCs$ would synergistically accelerate alkaline HOR. The reaction between surface adsorbed *OH and *H to form $H_2O$ is the rate-determining step of alkaline HOR (Fig. 4g). Energy barriers of this reaction are calculated to be 0.47, 0.82, and 1.53 eV on the surface of $Pt_6NCs$, $Pt_1SAs$, and PtNPs, respectively (Fig. 4h and Supplementary Fig. 31). The much lower energy barrier of $Pt_6NCs$ contributes to its remarkable HOR activity in alkaline media.

In terms of CO adsorption, DFT simulations reveal that due to the downshift of the $d$-band center based on the ligand effect, $Pt_6NCs$ possess a much lower CO adsorption strength ($-1.1$ eV) than that of PtNPs ($-1.46$ eV) and $Pt_6NCs$/C-550 ($-1.87$ eV) (Fig. 4i and Supplementary Fig. 32), suggesting its much lower binding energy to CO. These results explain the marked CO-tolerance ability of $Pt_6NCs$/C and agree well with above CO-stripping trend. Besides, stronger OHBE on $Pt_6NCs$ surface than PtNPs assists in the oxidation of adsorbed CO, which also leads to the enhanced CO-tolerance ability and recoverable catalytic activity.

In summary, we have successfully synthesized atomically precise $PPh_3$-stabilized $Pt_6NCs$ leveraging on creating a mild reduction environment and achieved superior HOR catalytic performance in the alkaline medium. Mechanism studies demonstrate that the $Pt_6NCs$ structure and $PPh_3$ ligand may play decisive roles in tuning the HBE, OHBE, $H_2O$ formation energy, and CO absorption energy. Benefiting from the self-optimized ligand effect and atomically precise six-Pt-atom structure, the $Pt_6NCs$/C catalyst exhibits much better HOR catalytic performance in terms of activity, durability, and CO-tolerant ability. This study exemplified the application of atomically precise Pt NCs for high-performance HOR catalysis, and the knowledge conveyed here may be inspirable for the future design of metal NCs-based electrocatalysts for energy conversion.

## Methods

**Materials and chemicals**. Chloroplatinic acid hexahydrate ($H_2PtCl_6\cdot6H_2O$, 37.5% Pt basis), triphenylphosphine ($PPh_3$, 99%), toluene (99.5%), borane-tert-butylamine complex (TBAB, ≥95.0%), sodium borohydride ($NaBH_4$, 98%), L-reduced glutathione (GSH, 98%), ethanol (≥99.5%), Nafion solution (~5% in a mixture of lower aliphatic alcohols and water), and commercial 20% Pt/C catalyst were purchased from Sigma-Aldrich. Carbon black (Vulcan XC-72R) was bought from Carbot Co. All reagents were used as received without further purification.

**Synthesis of $Pt_6NCs$**. In the first step, 200 μL of toluene containing 50 mM of $PPh_3$, 150 μL of 50 mM $H_2PtCl_6\cdot6H_2O$ aqueous solution, and 4.5 mL of ethanol were put into a beaker under continuous magnetic stirring. Then 100 μL of ethanol containing 100 mM of TBAB was added into the above solution and kept stirring for 3 h to obtain a dark brown Pt cluster solution. Finally, the Pt cluster sample could be collected for later use.

**Synthesis of $Pt_6NCs$/C**. 5 mL of Pt cluster suspension and 20 mg of XC-72R were successively added into 45 mL of ethanol and kept stirring for 6 h. The final product was collected by vacuum filtration and denoted as $Pt_6NCs$/C.

**Synthesis of $Pt_1SAs$/C**. In brief, 200 μL of toluene containing 50 mM of $PPh_3$, 200 μL of 50 mM $H_2PtCl_6\cdot6H_2O$ aqueous solution, and 50 mL of ethanol were added into a beaker under continuous magnetic stirring. After stirring for 30 min, 20 mg of XC-72R was added into the above suspension and kept stirring for 2 h. Finally, the final product was collected by vacuum filtration and denoted as $Pt_1SAs$/C.

**Synthesis of PtNPs/C**. 200 μL of 50 mM $H_2PtCl_6\cdot6H_2O$ aqueous solution was added to 50 mL of ethanol. 50 μL of ethanol containing 100 mM TBAB was introduced into the above mixture under agitated stirring. Next, 75 μL of toluene containing 50 mM of $PPh_3$ solution was added to form a protective layer of $PPh_3$ on the surface of Pt NPs. After stirring for 30 min, 20 mg of XC-72R was added into the above suspension and kept stirring for 2 h. Finally, the final product was collected by vacuum filtration and denoted as PtNPs/C.

**Synthesis of $Pt_6NCs$/C-550**. The control sample was synthesized by a similar process to that of $Pt_6NCs$/C except for the additional high-temperature calcination. $Pt_6NCs$/C catalyst was annealed under Ar at 550 °C for 5 h to form $Pt_6NCs$/C-550.

**Synthesis of PtNPs/C-$NaBH_4$**. 200 μL of 50 mM $H_2PtCl_6\cdot6H_2O$ aqueous solution was added to 40 mL of ethanol. Under continuous stirring, 10 mL of 15 mM $NaBH_4$ aqueous solution was introduced into the above solution. After stirring for 30 min, 20 mg of XC-72R was added into the above suspension and kept stirring for 30 min. Finally, the final product was collected by vacuum filtration and denoted as PtNPs/C-$NaBH_4$.

**Synthesis of GSH-$PtNCs$/C**. Typically, 5 mL of 50 mM $H_2PtCl_6\cdot6H_2O$ aqueous solution and GSH (0.94 mmol) were added into a beaker. 2.0 mL of $NaBH_4$ (1.9 mmol) was added dropwise under vigorous stirring. After stirring for 30 min, 20 mg of XC-72R was added into the above suspension and kept stirring for 2 h. Finally, the final product was collected by vacuum filtration and denoted as GSH-$PtNCs$/C.

**Physical characterizations**. UV-vis absorption spectra were recorded on an Agilent 8453 spectrometer. ESI-MS was conducted on a Bruker Impact II time-of-flight MS system. TEM images and HAADF-STEM images were obtained with a JEM-2100UHR field emission electron microscope at an accelerating voltage of 200 kV. Atomic resolution high magnification STEM image was obtained using an aberration-corrected transmission electron microscope (JEOL, JEM-ARM200F) operated at 200 kV. XRD patterns were collected on a D8 Advance X-ray powder diffractometer equipped with a Cu Kα radiation source (λ = 0.15405 nm) at 40 kV. All the diffraction data were collected in a 2θ range from 5° to 75° at a scanning rate of 8° $min^{-1}$. XPS analyses were acquired with a PHI 5000 Versa Probe spectrometer from ULVACPHI using an Al Kα (1486.6 eV) photon source. All binding energies were calibrated using the C(1s) carbon peak (284.8 eV), which was applied as an internal standard. TGA was performed on SDT650 from NETZSCH. The starting temperature was 20 °C with a 5 °C $min^{-1}$ ramp rate to 800 °C under 100 mL $min^{-1}$ airflow[55].

**Electrochemical measurements**. To prepare the electrode, 2.0 mg of catalyst, 490 μL of ethanol, and 10 μL of 5 wt.% Nafion solution were mixed and sonicated for 1 h to form a homogeneous catalyst ink. A certain amount of catalyst ink was dropped on the surface of the glassy carbon rotating disk electrode (RDE) in batches (diameter: 4 mm). For HOR, the loading was 5 μg $cm^{-2}$ (on the basis of the weight of Pt species in the whole experiment) to have a wide kinetic-diffusion mix controlled region so that Koutecky–Levich equation could be applied. Before any electrochemical measurement, three quick LSVs from $-0.02$ V to 0.5 V (vs. RHE) with a scan rate of 10 mV $s^{-1}$ were performed to activate the catalyst. 20 wt.% Pt/C was used as a reference and the loading amount was 10 μg $cm^{-2}$ on the RDE, and then dried naturally at room temperature. All electrochemical measurements were performed in a standard three-electrode system at room temperature by the RRDE-3A connected to CHI Electrochemical Station (Model 660d). All reference potentials have been converted to a reversible hydrogen electrode and were iR-corrected. The RDE coated with the catalyst, a Hg/HgO with 1.0 M KOH solution filled, and a graphite rod were employed as the working electrode, reference electrode, and counter electrode, respectively[55].

All electrochemical tests were performed in a 0.1 M KOH solution. The linear sweep voltammetry (LSV) was carried out in the KOH solution saturated with $H_2$ at various rotation rates from 400 to 3600 rpm to evaluate the catalytic activity of HOR. The test parameters of LSV were set to a voltage window from $-0.04$ V to 0.5 V (vs. RHE) and a sweep speed of 5 mV $s^{-1}$. The cyclic voltammetry (CV) curves were obtained in an Ar-saturated 0.1 M KOH solution by scanning the potential from 0.005 V to 1.1 V (vs. RHE) at 50 mV $s^{-1}$. The ECSAs were evaluated from the integral area of hydrogen underpotential deposition ($H_{upd}$) peaks with the subtraction of the double layer in the CV curve and a charge density of 210 μC $cm_{Pt}^{-2}$. Electrochemical impedance spectroscopy (EIS) measurement was conducted with a frequency ranging from 0.1 to $10^5$ Hz and an amplitude of 5 mV under an overpotential of 30 mV. The accelerated durability tests were carried out in 0.1 M KOH between 0.005 V and 0.6 V (vs. RHE) at a scan rate of 100 mV $s^{-1}$ for 2000 cycles. CO stripping was performed by holding the RDE at 0.1 V (vs. RHE) for 10 min in the purged CO to adsorb CO. Then RDE was quickly moved to a fresh 0.1 M KOH solution and recorded the two CV cycles in a potential region from 0 to 1.2 V (vs. RHE) at a sweep rate of 20 mV $s^{-1}$. The kinetic current density ($j_k$) can be obtained by the Koutecky–Levich (K–L) equation:

$$\frac{1}{j} = \frac{1}{j_k} + \frac{1}{j_d} = \frac{1}{j_k} + \frac{1}{BC_0\omega^{1/2}} \tag{1}$$

where $j$ is the measured current, which can be deconvoluted into $j_k$ and diffusional current ($j_d$) components; $B$ is the Levich constant; $C_0$ is the solubility of $H_2$ in the electrolyte; $\omega$ is the angular velocity of rotating disc electrode during measurements.

The exchange current density ($j_0$) can be calculated by fitting $j_k$ into the Butler-Volmer (B-V) equation:

$$j_k = j_0\left(e^{\frac{\alpha F}{RT}\eta} - e^{-\frac{(1-\alpha)F}{RT}\eta}\right) \tag{2}$$

where $\alpha$, $R$, $T$, and $\eta$ represent the transfer coefficient, the universal gas constant, the operating temperature, and the overpotential, respectively. In a small potential window of the micro-polarization region near the equilibrium potential, $j_k$ approximately equals to $j$. In this case, the B-V equation can be expanded by Taylor's formula and simplified as Eq. (3):

$$j_k = j_0\frac{\eta F}{RT} \tag{3}$$

By linearly fitting the polarization curve in the micro-polarization region, the $j_0$ can be obtained.

**Computational models and methods**. To comprehensively consider the accuracy and efficiency of the calculations, the $P(C_6H_5)_3$ ligand was represented by the $P(CH_3)_3$ group in this work. Therefore, according to the experimental results, the $Pt_1$ species was built by a Pt coordinated by two $P(CH_3)_3$ groups ($Pt(P(CH_3)_3)_2$), while the $Pt_6$ species was modeled as a $Pt_6$ cluster coordinated by five Cl ions and four $P(CH_3)_3$ groups ($Pt_6(P(CH_3)_3)_4Cl_5$). To study PtNPs, Pt(1 1 1) was built using a periodic four-layer slab with a p(4 × 4) unit cell. To study $Pt_6$NCs/C-550, the Pt octahedron was constructed by using six Pt atoms. A vacuum of 20 Å in thickness was used to separate the catalyst surface from its periodic image in the direction along the surface normal.

All the calculations were performed by the spin-polarized density functional theory (DFT) method. Each model was optimized by minimizing the energy by the DFT calculation. The metal ion cores were treated by density functional semi-core pseudopotential (DSPP)[56], while the valence electrons were represented by the double numerical plus polarization (DNP) basis set. The exchange-correlation energy was calculated by the Perdew-Burke-Ernzerhof (PBE) generalized gradient approximation[57]. Moreover, the Grimme' PBE+D2 correction was adopted to consider the long-range dispersion forces, which exhibits comparable accuracy with the PBE0 hybrid XC functional including an ab initio van der Waals correction[58,59]. According to the Monkhorst-Pack (MP) scheme, the Brillouin zone was sampled with a 4 × 4 × 1 k-mesh for the slab calculation. The convergence tolerance of the geometry optimization was set as $1 \times 10^{-5}$ Ha for energy, $2 \times 10^{-3}$ Ha Å$^{-1}$ for maximum force, and $5 \times 10^{-3}$ Å for maximum displacement[60–62]. The transition state was searched by the linear synchronous transition/quadratic synchronous transit (LST/QST) method, confirmed by the frequency calculations. The geometry optimization, transition state search, and frequency calculations were performed by the DMol$^3$ code[59,61].

The density of states (DOS) and electron density difference were calculated by the CASTEP code with a cutoff energy of 750 eV[63]. In calculations, the norm-conserving pseudopotential was chosen[64]. The Brillouin zone was sampled using 8 × 8 × 8 k-points. The other details were consistent with the above DMol$^3$ computations.

The change of free energy for the reaction was calculated by Eq. (4):

$$\Delta G = \Delta E + \Delta ZPE - T\Delta S \tag{4}$$

where $\Delta E$, $\Delta ZPE$, and $\Delta S$ are the change of electronic energy, zero-point energy, and entropy at the temperature of 298 K, respectively.

The adsorption energy of an adsorbate ($E_{ads}$) was calculated by Eq. (5):

$$E_{ads} = E_{adsorbate+sub} - E_{adsorbate} - E_{sub} \tag{5}$$

where $E_{adsorbate+sub}$ is the total energy of the adsorbed system, and $E_{adsorbate}$ is the energy of an adsorbate.

The d-band center ($\varepsilon_d$) of the catalysts was calculated by Eq. (6):

$$\varepsilon_d = \frac{\int_{-\infty}^{+\infty} E\rho_d(E)dE}{\int_{-\infty}^{+\infty} \rho_d(E)dE} \tag{6}$$

where $\rho_d(E)$ is the density of d states at an energy level E.

## Data availability

Source data are provided with this paper, which can also be available from the corresponding authors on reasonable request. Additionally, data reported herein have been deposited in the Figshare database, and are accessible through https://doi.org/10.6084/m9.figshare.19241946. Source data are provided with this paper.

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

## Acknowledgements

This work was supported by the National Natural Science Foundation of China (51877216, 21805307, 21905300, and 22071127), Taishan Scholar Foundation (tsqn20161017 and tsqn201812074), the Natural Science Foundation of Shandong Province (ZR2019YQ07), Fundamental Research Funds for the Central Universities (21CX06011A), and Graduate Student Innovation Projects of China University of Petroleum (YCX2021097).

## Author contributions

W.X. and X.Y. conceived and supervised the project. X.N.W. and X.J.L. performed the experiments, collected and analyzed the data. L.M.Z. carried out the DFT calculations. Y.L., Y.S.W., Q.F.Y., J.P.X., Q.Z.X., and Z.F.Y. helped with electrochemical data collection and analysis. X.N.W. and X.Y. co-wrote the manuscript. All authors discussed the results and commented on the manuscript.

## Competing interests

The authors declare no competing interests.
