## [Peer Review File · Nature Communications]

Atomic-precision Pt₆ nanoclusters for enhanced hydrogen electro-oxidationREVIEWER COMMENTS

Reviewer #1 (Remarks to the Author):

In this paper, the authors reported the preparation of Pt₆ nanoclusters and their enhanced catalytic ability towards the hydrogen electro-oxidation. The Pt₆ nanocluster was co-protected by PPh₃ and Cl ligands, and its composition was confirmed by the ESI measurement. The Pt₆/C electrocatalyst delivers the remarkable CO-tolerant ability that conventional Pt/C catalyst lacks. Calculation results demonstrated that the enhanced electrocatalytic performance could be attributable to the bifold effects of the PPh₃ ligand. However, some necessary discussions and experiments are lacking in its current form. No precise structure is involved in, and there is something imprecise towards the mass analysis. Therefore, I cannot recommend the publication of this paper at the actual state. (1) In Figure 1C, the experimentally obtained isotope pattern of Pt₆(PPh₃)₄Cl₅ is in NOT good agreement with the simulated one, and 1-2 Da gap was there. Some explanation should be added. And the following discussions towards the nanocluster structure should be presented more carefully since the structure was not experimentally solved. (2) The mass results only presented the signal within a range from 1800 to 3000 Da. And no purification process was performed before the loaded onto C. In this context, it is not rigorous to conclude that the catalytic agent is Pt₆/C. There might be other Pt complexes in this reaction system. Using the Pt₆ crystal in this process can eliminate these interferences. (3) In Methods, the preparation of Pt₆ nanoclusters was absent. How about the stability of the nanocluster before its loading on C.

Reviewer #2 (Remarks to the Author):

The work by Xing and co-workers is a nice discussion on the activity of Pt-NC starting from unprecedented synthesis and characterization and followed by r electrocatalytic hydrogen oxidation reaction (HOR), of interest in current technological needs. The work is well carried out, and the discussion is easy to follow. Hence the manuscript is proposed for acceptance in Nature Communications after considering the following comments:

- 1) The HOR measurements were obtained at 0.1 M KOH. How relevant is the pH for this activity? i.e 0.05 M KOH, etc...
- 2) The activity of the cluster is ascribed firstly to the structure of the cluster. However, no further discussion of the "active sites" is included, which can be obtained from calculations, for example, from electrostatic potential surfaces as depicted by Brinck in the following paper: <https://pubs.acs.org/doi/10.1021/jacs.7b05987>
- 3) The electronic structure is discussed in terms of the overall d-band position. How can this characteristic be further tuned?, for example, by envisaging N-Heterocyclic carbene ligands? This effect has been discussed for gold, for example, in: <https://doi.org/10.1039/C9QI00513G>

Reviewer #3 (Remarks to the Author):

This work reports a highly active Pt₆ cluster as an electrocatalyst for H₂ oxidation reaction (HOR). The HOR performance is much better than that of the single-atom Pt₁ and also regular PtNPs. The PPh₃ ligands on Pt₆ are concluded to play a crucial role in the catalytic activity and anti-CO-poisoning, which is attributed to the weakened binding of *H and *CO

on Pt6 and enhanced adsorption of *OH, all benefiting the HOR.

The results are potentially interesting, and the work is quite thorough. Some technical questions are as follows.

1. Pt6 should be much smaller than the observed size of 1-1.2 nm (Fig 1e). Thus, the ESI and TEM data seem inconsistent.
2. The presence of PPh3 on Pt6 is concluded to be critical for the high HOR activity and durability. But Pt6 is protected by both PPh3 and Cl (five) ligands. What about the effect of Cl other than PPh3?
3. The authors discussed *OH reaction with CO for the removal. Would *OH also remove PPh3 and Cl on Pt6 (hence, partially deprotected Pt6 and better activity)? While the fully-deprotected Pt6NCs/C-550 is demonstrated to be not good, what about a brief thermal treatment to partially deprotect Pt6?
4. On the CO poisoning, it is related to the question of Pt-PPh3 binding strength. Typically, PPh3 should be stronger than CO in binding, but the authors' results seem to indicate that PPh3 is weaker than CO. Can the authors compare the binding energy of PPh3 and CO on the Pt6?
5. In Suppl Fig 26, did the various sites of Pt6 show similar adsorption energy for *H or is there a specific site for an optimum H-adsorption?

Reviewer #4 (Remarks to the Author):

Wang et al reported the synthesis, characterization, and HOR activity of Pt nanocluster decorated by PPh3 ligand. In my view, the mechanism of HOR activity is well demonstrated with sufficient experimental data and calculation. I would like to recommend the publication of this article to Nature Communications after the authors address some minor issues:

1. Is the coordination structure with the PPh3 ligand still stable after HOR stability test? The MS spectra or other methods should be added for identification.
2. Scale bar for charge density should be shown in Fig 6c. Two values only referring to red and blue colors cannot cover the complete information of this figure.
3. In Fig 4d, PDOS of d orbital should also be given.

Response to Reviewers' Comments and Revisions Made

Response to Reviewer #1's Comments:

In this paper, the authors reported the preparation of Pt₆ nanoclusters and their enhanced catalytic ability towards the hydrogen electro-oxidation. The Pt₆ nanocluster was co-protected by PPh₃ and Cl ligands, and its composition was confirmed by the ESI measurement. The Pt₆/C electrocatalyst delivers the remarkable CO-tolerant ability that conventional Pt/C catalyst lacks. Calculation results demonstrated that the enhanced electrocatalytic performance could be attributable to the bifold effects of the PPh₃ ligand. However, some necessary discussions and experiments are lacking in its current form. No precise structure is involved in, and there is something imprecise towards the mass analysis. Therefore, I cannot recommend the publication of this paper at the actual state.

Reply: We appreciate the reviewer for his/her thoughtful comments and questions. To further improve the quality of this manuscript as well as address your concerns about the structural precision of Pt₆ clusters, we have tried our best to revise the manuscript according to your valuable suggestions.

(1) In Figure 1C, the experimentally obtained isotope pattern of Pt₆(PPh₃)₄Cl₅ is in NOT good agreement with the simulated one, and 1-2 Da gap was there. Some explanation should be added. And the following discussions towards the nanocluster structure should be presented more carefully since the structure was not experimentally solved.

Reply: We highly appreciate the reviewer's perceptive comments. Upon careful analysis, we realized that the 1-2 Da gap in the molecular weight is due to the overdue calibration of the ESI-MS equipment. As shown in **Figure R1**, by comparing the isotope patterns of the Pt₆(PPh₃)₄Cl₅ NCs before and after calibration with the simulated one, it is found that the isotope pattern of the Pt NCs with proper calibration matches nicely with the simulated one. We are sorry for this stupid mistake made and thankful for the reviewer's timely reminder in this issue. In this revision, we have revised **Figure 1c** by updating the calibrated data.

Figure R1. Three isotope patterns of the Pt₆(PPh₃)₄Cl₅ NCs: experimentally acquired ones before (black curve) and after (blue curve) ESI-MS calibration, and simulated one (red curve).

In addition, we totally agree with the reviewer's viewpoint on the discussion towards nanocluster structure. At the current state, we are unable to acquire the crystal structure of the Pt₆NCs by single-

crystal X-ray crystallography because it requires very sophisticated skills in single crystal growth that is out of our expertise. Therefore, we employed the DFT simulation to provide some structural information about the Pt₆NCs. According to the reviewer's valuable suggestion, we have revised the manuscript with careful discussion on the NC structure in this revision.

Revision:

Figure 1c, Page 7

Figure 1c ESI-MS spectrum of Pt₆NCs (measured in positive mode); inset: (left) experimentally obtained (green curve) and simulated (black curve) isotope patterns of Pt₆(PPh₃)₄Cl₅; (right) a simulated structure model of a Pt₆NC.

Caption of Figure 4b-c, Page 18

“**b** Simulated model of Pt₆NCs (left) and electron density differences of Pt₆NCs (right), where the blue and yellow represent accumulation and depletion of charge, respectively. **c** The slice perspective of differential charge density distribution for Pt₆NCs based on DFT analysis.”

Paragraph 1, Page 19

“However, it should be mentioned that the structures of the Pt₆NCs as well as Pt₁SAs and PtNPs discussed here are simulated based on DFT, and their actual structures still rely on the single-crystal X-ray crystallography, which will be realized with the future research efforts. Nevertheless, the DFT simulation could provide useful information to deepen our fundamental understanding in this issue. Of note, based on DFT analysis, ...”

Paragraph 4, Page 22

“Mechanism studies demonstrate that the Pt₆NCs structure and PPh₃ ligand may play decisive roles in tuning the HBE, OHBE, H₂O formation energy, and CO absorption energy.”

(2) The mass results only presented the signal within a range from 1800 to 3000 Da. And no purification process was performed before the loaded onto C. In this context, it is not rigorous to conclude that the catalytic agent is Pt₆/C. There might be other Pt complexes in this reaction system. Using the Pt₆ crystal in this process can eliminate these interferences.

Reply: Thank you so much for the reviewer's constructive comments. The reviewer is concerned about the correlation between the size of Pt NCs and the electrocatalytic activity. Therefore, the first thing is the identification of the Pt species in the solution of Pt₆NCs in order to well address this concern. As shown in **Figure R2a**, there is a single peak in the ESI mass spectrum within the range from 1800 to 10000 Da, which suggests that no larger Pt NCs could be formed in the synthetic

system, demonstrating the good monodispersity of the Pt NCs. However, as the reviewer mentioned, without purification, there are indeed some unreacted complexes in the sample solution, which is supported by the ESI-MS result (**Figure R2b**). As shown in **Figure R2b**, in total, 9 complex species are identified, and their formulas are listed in **Figure R2c**. To estimate the influence of these Pt complexes on the electrocatalytic activity, we prepared a reference sample merely containing these Pt complexes but without the Pt₆NCs, and subjected it to electrocatalytic tests. As shown in **Figure R3**, the reference sample has 11 complex species in total, which generally covers the population of the complexes in the Pt₆NCs solution (**Figure R2b**).

Figure R2. ESI-MS spectrum of Pt₆NCs (measured in positive mode) in the range of 1800-10000 Da (a), and in the range of 380-1800 Da (b). (c) Molecular formulas of the labeled species.

Figure R3. ESI-MS spectrum of the reference sample (measured in positive mode) in the range of 380-1800 Da (a), and (b) molecular formulas of the labeled species. Note: The reference sample is synthesized as follows. In the first step, 200 μ L of toluene containing 50 mM of PPh₃, 200 μ L of 50 mM H₂PtCl₆·6H₂O aqueous solution, and 4.5 mL of ethanol were put into a beaker under continuous magnetic stirring. Then 25 μ L of ethanol containing 100 mM of TBAB was added into the above solution and kept stirring for 10 min. Finally, the reference sample could be collected for later use.

The HOR activity of unpurified Pt₆NCs and reference sample catalysts was evaluated in H₂-saturated 0.1 M KOH aqueous solution. The result reveals that the j_0 of the unpurified Pt₆NCs is 7.5 times higher than that of the reference sample (3.23 mA cm_{disk}⁻² vs. 0.43 mA cm_{disk}⁻²) (see **Figure R4**), which corroborates that the outstanding electrocatalytic activity is predominately contributed by the Pt₆NCs rather than those Pt complex species. Here we are sorry that we are unable to crystallize the Pt₆NCs because it requires very sophisticated skills in single crystal growth that is out

of our expertise, and we hope the above analysis as well as the control experiment we conducted could satisfy the reviewer in this issue.

Figure R4. HOR performance of Pt₆NCs/C and reference sample containing Pt complexes (deposited on the same carbon support).

(3) In Methods, the preparation of Pt₆ nanoclusters was absent. How about the stability of the nanocluster before its loading on C.

Reply: We are grateful for the reviewer’s valuable comments. According to the reviewer’s suggestion, we have supplemented the synthetic procedure of Pt₆NCs to the Methods Section. In addition, the stability of the Pt₆NCs is quite good. As shown in **Figure R5**, the Pt₆NCs could be maintained after storage of three weeks at 4 °C without N₂ protection (Note: the stability test is ongoing), which is evidenced by the unchanged peak of Pt₆NCs in the ESI-MS spectra. In this revision, we have included **Figure R5** in the Supplementary Information as **Supplementary Figure 5**, and added one sentence into the context to discuss the stability of the Pt₆NCs.

Figure R5. ESI-MS spectra of the Pt₆NCs freshly prepared (blue curve) and after 3 weeks storage (red curve) at 4 °C without N₂ protection.

Revision:

Methods Section, Page 23

“**Synthesis of Pt₆NCs.** In the first step, 200 μL of toluene containing 50 mM of PPh₃, 150 μL of 50 mM H₂PtCl₆·6H₂O aqueous solution, and 4.5 mL of ethanol were put into a beaker under continuous magnetic stirring. Then 100 μL of ethanol containing 100 mM of TBAB was added into the above solution and kept stirring for 3 h to obtain a dark brown Pt cluster solution. Finally, the Pt cluster sample could be collected for later use.”

Paragraph 1, Page 8

“In addition, the as-fabricated Pt₆NCs display good stability, and there is no obvious change in the ESI-MS spectra for the Pt₆NCs after three weeks storage at 4 °C without N₂ protection (Supplementary Fig. 5).”

Supplementary Figure 5, Page 3 in Supplementary Information

Supplementary Figure 5. ESI-MS spectra of the Pt₆NCs freshly prepared (blue curve) and after 3 weeks storage (red curve) at 4 °C without N₂ protection.

Response to Reviewer #2's Comments:

The work by Xing and co-workers is a nice discussion on the activity of Pt-NC starting from unprecedented synthesis and characterization and followed by electrocatalytic hydrogen oxidation reaction (HOR), of interest in current technological needs. The work is well carried out, and the discussion is easy to follow. Hence the manuscript is proposed for acceptance in Nature Communications after considering the following comments:

Reply: We greatly appreciate the reviewer's acknowledgment in the significance, novelty, and broad interest of our manuscript. We also believe that this manuscript could shed light on the design of other high-performance Pt-based electrocatalysts, and this work may quickly show its impact in multi-disciplinary fields.

1) The HOR measurements were obtained at 0.1 M KOH. How relevant is the pH for this activity? i.e 0.05 M KOH, etc...

Reply: Many thanks for the reviewer's valuable comment and suggestion. Following the reviewer's suggestion, we performed HOR activity tests in H₂-saturated KOH electrolytes with different concentrations (0.05 M, 0.1 M, 0.2 M, and 0.5 M). As shown in **Figure R6a**, the current response speed increases with increasing the pH in the kinetic and diffusion mixed controlled regions (< 0.1 V, vs. RHE), indicating a positive correlation between activity and pH in alkaline environments. In addition, the half-wave potential is often employed as the activity indicator in the field of electrocatalysis. It is noted that the half-wave potential should not be treated as the pure kinetic parameter as it includes both kinetic overpotential and H₂ diffusion overpotential due to the limitation using the RDE method, where H₂ mass transport is constrained (*Nat. Commun.* **2015**, *6*, 5848). However, as shown in **Figure R6b**, by normalizing the HOR currents to their limiting currents in different electrolytes, HOR in each electrolyte has the same diffusion overpotential. Here the trend generated using half-wave potential as the activity indicator in this pH regime is valid as

it contains the same diffusion overpotential, thus clearly demonstrating the correlation between the reaction kinetics changes and the pH. On the basis, it is found that the half-wave potentials are 33.3, 22.9, 16.8, and 13.1 mV for Pt₆NCs/C in 0.05 M KOH solution (pH ~12.7), 0.1 M KOH solution (pH ~13.0), 0.2 M KOH solution (pH ~13.3), and 0.5 M KOH solution (pH ~13.7), respectively, suggesting the increased reaction rates with elevating the pH.

Figure R6. (a) HOR polarization curves of Pt₆NCs/C collected in H₂-saturated KOH aqueous solutions at different pH. The sweep rate is 5 mV s⁻¹ and the rotating speed is 2500 rpm. Note that the polarization curves here are not corrected with solution resistance. (b) HOR polarization curves of Pt₆NCs/C normalized to the maximum limiting current density (j_{lim}).

In addition, the response of HOR activity to all pH conditions has also been reported in some excellent papers (*Nat. Chem.* **2013**, 5, 300; *ACS Catal.* **2019**, 9, 6194; *Sci. Adv.* **2016**, 2, e1501602), which is in good agreement with our experimental results obtained under alkaline conditions. It is important to note that most of the reported works on alkaline HOR catalysis employed 0.1 M KOH aqueous solution as the electrolyte (*Energy Environ. Sci.* **2021**, 14, 2620; *Nano Energy* **2018**, 44, 288) by comprehensively considering the performance balance, practicality, and cost. Given that the relationship between pH and activity is not the focus of this work, we, therefore, employed the universal 0.1 M of KOH aqueous solution as the electrolyte in order to obtain experimental results that can be compared with most of the works.

2) The activity of the cluster is ascribed firstly to the structure of the cluster. However, no further discussion of the "active sites" is included, which can be obtained from calculations, for example, from electrostatic potential surfaces as depicted by Brinck in the following paper: <https://pubs.acs.org/doi/10.1021/jacs.7b05987>

Reply: We highly appreciate the reviewer's great effort in further improving the quality of this manuscript. According to the reviewer's instructive suggestion, we employed the surface electrostatic potential to discuss the catalytically active sites of the Pt₆NCs catalysts.

Figure R7 shows the surface electrostatic potential mapped onto the total electron density of the bare Pt₆NCs and ligand-protected Pt₆NCs. The red regions of the molecular surface exhibit a low electrostatic potential, indicating nucleophilic properties, while the high electrostatic potential (blue color) of the molecular surface suggests the electrophilic properties. The electrostatic potential surface for bare Pt₆NCs confirms the higher electron affinities of the Pt atoms at the tetrahedral vertices (**Figure R7a**). The positive electrostatic potential represented by the dark blue regions delimits electrophilic zones, where the cluster is susceptible to chemical bonding by nucleophiles

such as PPh₃ and Cl. As discussed above, apical Pt sites are the most stable sites for PPh₃ and Cl binding which provide strong Pt interaction and stabilize the cluster geometry.

Figure R7. Surface electrostatic potentials of (a) bare Pt₆ crystal and (b) Pt₆NCs. Surface electrostatic potential mapped on the isosurface (0.0001 a.u.) of electronic density.

When the geometry of the Pt₆NCs is maintained, the charges coming from the unshared pair of electrons of PPh₃ distribute over the Pt core, and a slight charge redistribution occurs at Pt-P bonding, creating nucleophilic areas at the center sites of the cluster, as shown by the red regions of the electrostatic potential of Pt₆NCs in **Figure R7b**. In this red negative potential region, electrophiles or positively charged species (such as hydrogen intermediates) can approach the cluster core. On the other hand, Pt₆NCs present a limited negative electrostatic potential sterically obstructed by ligands bonded to Pt atoms, resulting in a reduced desorption energy barrier for *H in the Volmer step. Correspondingly, DFT calculations disclose that Pt₆NCs catalyst yields a ΔG_{*H} of -0.07 eV that is very close to the ideal value ($\Delta G_{*H} = 0$) for HOR.

In this revision, we have supplemented this **Figure R7** into the Supplementary Information as **Supplementary Figure 28**, and also added corresponding discussions to both the context and Supplementary Information (**Supplementary Note V**) to clarify this issue. Moreover, the interesting reference that the reviewer mentioned was also cited in the manuscript as Ref. 50 to support our discussion.

Revision:

Paragraph 1, Page 20

“In addition, it is found the Gibbs free energy of *H (ΔG_{*H} for short) of Pt₆NCs (-0.07 eV; Fig. 4e) on the optimal adsorption site (Pt-P bonding) is very close to the ideal value ($\Delta G_{*H} = 0$) for HOR (Supplementary Fig. 27).”

“In parallel, the formation mechanism and tunable nucleophilicity of Pt₆NCs can be well elucidated by the limited surface negative electrostatic potential obstructed by ligands bonded to Pt atoms (Supplementary Fig. 28)⁵⁰, consistent with the DFT analysis.”

Reference 50, Page 29

“50. Stenlid JH, Brinck T. Extending the σ -hole concept to metals: an electrostatic interpretation of the effects of nanostructure in gold and platinum catalysis. *J. Am. Chem. Soc.* **139**, 11012-11015 (2017).”

Supplementary Figure 28, Page 17 in Supplementary Information

Supplementary Figure 28. Surface electrostatic potentials of (a) a bare Pt₆NC and (b) ligand-protected Pt₆NCs. Surface electrostatic potential mapped on the isosurface (0.0001 a.u.) of electronic density.

Supplementary Note V, Supplementary Information

“**Supplementary Note V:** Supplementary Figure 28 shows the surface electrostatic potential mapped onto the total electron density of the bare Pt₆NCs and ligand-protected Pt₆NCs. The red regions of the molecular surface exhibit a low electrostatic potential, indicating nucleophilic properties, while the high electrostatic potential (blue color) of the molecular surface suggests the electrophilic properties. The electrostatic potential surface for bare Pt₆NCs confirms the higher electron affinities of the Pt atoms at the tetrahedral vertices (Supplementary Figure 28a). The positive electrostatic potential represented by the dark blue regions delimits electrophilic zones, where the cluster is susceptible to chemical bonding by nucleophiles such as PPh₃ and Cl. As discussed above, apical Pt sites are the most stable sites for PPh₃ and Cl binding which provide strong Pt interaction and stabilize the geometry. When the geometry of the Pt₆NCs is maintained, the charges coming from the unshared pair of electrons of PPh₃ distribute over the Pt core, and a slight charge redistribution occurs at Pt-P bonding, creating nucleophilic areas at the center sites of the cluster, as shown by the red regions of the electrostatic potential of Pt₆NCs in Supplementary Figure 28b. In this red negative potential region, electrophiles or positively charged species (such as hydrogen intermediates) can approach the cluster core. On the other hand, Pt₆NCs present a limited negative electrostatic potential sterically obstructed by ligands bonded to Pt atoms, resulting in a reduced desorption energy barrier for *H in the Volmer step.”

3) *The electronic structure is discussed in terms of the overall d-band position. How can this characteristic be further tuned?, for example, by envisaging N-Heterocyclic carbene ligands? This effect has been discussed for gold, for example, in: <https://doi.org/10.1039/C9QI00513G>*

Reply: We are grateful for the reviewer’s unique research perspective in tuning the overall d-band position of the Pt₆NCs. As the reviewer expected, the N-heterocyclic carbenes (NHC) could be employed to further tune the d-band position of the Pt₆NCs according to the DFT analysis. In this study, based on the molecular structure of Pt₆(PPh₃)₄Cl₅, we constructed three Pt₆(NHC)₄Cl₅ models by replacing PPh₃ ligands with different NHC ligands to study the electronic regulation ability of the ligands (**Figure R8a-c**). We found that the incorporation of electron-donor or -withdrawing NHC ligands may strongly influence the molecular properties and tune the electronic structure via ligand engineering. For the HOR catalysis, different ligands significantly modulate the d-band

center of Pt, leading to the possibility of optimizing the HBE (**Figure R8d, e**). This electronic regulation strategy may provide a new research direction for the electronic regulation of catalysts for alkaline HOR catalysis, which is very interesting and could be our focus of the next research project although the synthesis of NHC-protected Pt NCs may be a grand challenge for the nanocluster community.

In this revision, we have supplemented **Figure R8** into the Supplementary Information as **Supplementary Figure 29**, and included several sentences in the manuscript to discuss it. Moreover, the interesting work mentioned by the reviewer has also been cited in the manuscript as Ref. 51.

Figure R8. (a-c) Structures of different N-heterocyclic carbene (NHC) ligand-protected Pt₆ NCs, and their NHC ligands. (d) The PDOSs of Pt 5d in 1#, 2#, and 3# models (each *d*-band center is marked by a dashed line) with the Fermi level aligned at 0 eV. (e) ΔG*_H on 1#, 2#, and 3# models.

Revision:

Paragraph 1, Page 20

“Indeed, the attribute of the rich surface chemistry of metal NCs may offer broad flexibility in regulating the *d*-band center of Pt NCs. For example, DFT simulation reveals that different N-heterocyclic carbene-derivative ligands ranging from strong to weak σ -donors can be employed to tune the *d*-band center of the Pt₆NCs, thereby regulating the HBE (Supplementary Fig. 29)⁵¹. This strategy may provide a new research direction in the electronic regulation of catalysts for alkaline HOR catalysis.”

Reference 51, Page 29

“51. Munoz-Castro A. Potential of N-heterocyclic carbene derivatives from Au₁₃(dppe)₅Cl₂ gold superatoms. Evaluation of electronic, optical and chiroptical properties from relativistic DFT. *Inorg. Chem. Front.* **6**, 2349-2358 (2019).”

Supplementary Figure 29, Page 18 in Supplementary Information

Supplementary Figure 29. (a-c) Structures of different N-heterocyclic carbene (NHC) ligand-protected Pt₆ NCs, and their NHC ligands. (d) The PDOSs of Pt 5d in 1#, 2#, and 3# models (each *d*-band center is marked by a dashed line) with the Fermi level aligned at 0 eV. (e) ΔG_{*H} on 1#, 2#, and 3# models.

Response to Reviewer #3's Comments:

*This work reports a highly active Pt₆ cluster as an electrocatalyst for H₂ oxidation reaction (HOR). The HOR performance is much better than that of the single-atom Pt₁ and also regular PtNPs. The PPh₃ ligands on Pt₆ are concluded to play a crucial role in the catalytic activity and anti-CO-poisoning, which is attributed to the weakened binding of *H and *CO on Pt₆ and enhanced adsorption of *OH, all benefiting the HOR. The results are potentially interesting, and the work is quite thorough. Some technical questions are as follows.*

Reply: We greatly appreciate the reviewer for the positive feedback on the content of this manuscript. The issues raised by the reviewer and the valuable comments have been elaborately solved in this revision, and please see more details below.

1. Pt₆ should be much smaller than the observed size of 1-1.2 nm (Fig 1e). Thus, the ESI and TEM data seem inconsistent.

Reply: We are thankful for the reviewer's valuable comment, and totally agree with the reviewer's viewpoint on this issue. Firstly, we performed DFT analysis to estimate the size of the most stable Pt₆(PPh₃)₄Cl₅ structure (**Figure R9**). As shown, it is found that the core diameter of the octahedral Pt₆ crystal is ~3.8 Å, while the distance of the para-position PPh₃ ligand is ~8.1 Å, and the size of the whole cluster reaches ~12.9 Å. As the reviewer mentioned, the diameter of the unprotected Pt₆ core should be less than 1 nm, but the size of the clusters clearly increases in the presence of multiple ligands. Of course, it is known that although small molecule ligands composed of lower atomic-number elements may be difficult to be detected by HAADF-STEM, this possibility does not seem

to be completely ruled out. Similar phenomena can also be observed in some ligand-protected metal clusters, such as $\text{Au}_4\text{Pt}_2(\text{SR})_8$ (~2 nm) (*Nat. Commun.* **2020**, 11, 4389) and $[\text{Pt}(\text{SR})_2]_6$ (~0.8 nm) (*Nat. Commun.* **2017**, 8, 688). We infer that several reasons may cause the observed cluster size to exceed the theoretical value: (1) The process of loading onto the carbon support may lead to the physical stacking and agglomeration of the clusters; (2) The high-energy electron beam emitted by STEM causes the fusion and subsequent flattening of Pt_6NCs , which is a usual phenomenon when using TEM to characterize the size of such tiny metal NCs (*J. Am. Chem. Soc.* **2009**, 131, 16672). It should be noted that TEM that is often used to characterize metal nanoparticles is less capable of characterizing the accurate size of ultrasmall metal NCs (*RSC Adv.*, **2014**, 4, 60581), and that is why the research community usually employs mass spectrometry to analyze the size or molecular weight of metal NCs (*Angew. Chem. Int. Ed.* **2019**, 58, 11967). Taken together, we believe that the slightly larger size of Pt_6NCs we observed is within an acceptable range and we hope the above analysis could satisfy the reviewer in this issue.

Figure R9. Simulated structure model of $\text{Pt}_6(\text{PPh}_3)_4\text{Cl}_5$.

2. The presence of PPh_3 on Pt_6 is concluded to be critical for the high HOR activity and durability. But Pt_6 is protected by both PPh_3 and Cl (five) ligands. What about the effect of Cl other than PPh_3 ?

Reply: Thank you for the insightful question. Owing to the difficulties in distinguishing the effects of Cl ligands and PPh_3 ligands alone on HOR activity through an experimental approach, we resorted to DFT simulations to distinguish the contributions of Cl ligands and PPh_3 ligands to HOR activity. In the Pt core of Pt_6NCs , there are three types of Pt atoms with different coordination environments, namely (1) Pt-P, (2) Pt-Cl, and (3) P-Pt-Cl (**Figure R10**). We investigated the origin of the high HOR activity by comparing the *d*-band centers, HBE, and OHBE of the different Pt sites described above.

Figure R10. Molecular structure of Pt₆NCs with different sites of Pt atoms. (1) Pt-P, (2) Pt-Cl, and (3) P-Pt-Cl.

It is found that the ΔG_{*H} of site 1 (-0.07 eV) is very close to the ideal value ($\Delta G_{*H} = 0$) for HOR. In comparison, the ΔG_{*H} values of site 2 (+0.24 eV) and site 3 (+0.46 eV) demonstrate larger deviation from the ideal value (**Table R1**). These results indicate that the PPh₃ ligands can significantly optimize the HBE of Pt compared to the Cl ligands. Since the *OH plays a crucial role to remove the *H in the Heyrovsky and Volmer steps, increasing *OH coverage on Pt would greatly benefit alkaline HOR. In addition, site 1 is found to possess a much more negative ΔG_{*OH} (-2.57 eV) than those of site 2 (-2.07 eV) and site 3 (-1.85 eV) due to the upshifted *d*-band center (**Figure R11**), suggesting its enhanced adsorptions of *OH. Based on the above results, we may conclude that the Pt sites bridged with PPh₃ ligands possess better HBE and OHBE than those with Cl ligands, leading to higher HOR activity.

Table R1. Comparison of the *d*-band centers, ΔG_{*H} , and ΔG_{*OH} on three types of Pt sites of Pt₆NCs.

	Site 1	Site 2	Site 3
d -band center	-2.842	-2.973	-3.400
ΔG_{*H}	-0.07	+0.24	+0.46
ΔG_{*OH}	-2.57	-2.07	-1.85

Figure R11. The PDOSs of Pt 5*d* of the Pt₆NCs in different sites.

To elucidate the effect of Cl ligands on catalytic activity, we simulated a Pt₆NC that is merely protected with Cl ligands instead of PPh₃ ligands (**Figure R12**). As shown, the Cl-protected Pt₆NC model contains only five Cl ligands with uncoordinated A sites and Cl-coordinated B sites on the surface. In this case, the ΔG_{*H} and ΔG_{*OH} of both A and B sites are inferior to Pt of Pt-P bonding in Pt₆NCs model (**Table R2**), strongly suggesting that the high activity originates from Pt-PPh₃ configuration rather than Pt-Cl.

Figure R12. Hypothetical model of Pt₆ NCs merely protected by Cl ligands.

Table R2. Comparison of the HBE and OHBE at A site and B site.

	Site A	Site B
ΔG_{*H}	-0.47	-0.38
ΔG_{*OH}	-2.88	-2.29

In summary, we have added these new data and corresponding discussion in the revised manuscript (**Paragraph 1, Page 20**) and Supplementary Information (**Supplementary Figure 27**) to clarify this aspect.

Revision:

Paragraph 1, Page 20

“In addition, it is found that the Gibbs free energy of ^{*}H (ΔG_{*H} for short) of Pt₆NCs (-0.07 eV; Fig. 4e) on the optimal adsorption site (Pt-P bonding) is very close to the ideal value ($\Delta G_{*H} = 0$) for HOR (Supplementary Fig. 27).”

Supplementary Figure 27, Page 17 in Supplementary Information

Supplementary Figure 27. Models of H adsorption on catalysts. (a) Pt₆NCs. (b) Pt₁SAs. (c) PtNPs. The gray, white, blue, pink, and reseda spheres represent C, H, Pt, P, and Cl atoms, respectively.

Supplementary Note IV, Page 17 in Supplementary Information

“**Supplementary Note IV:** Owing to the difficulties in distinguishing the effects of Cl ligands and PPh₃ ligands alone on HOR activity through an experimental approach, we resorted to DFT simulations to distinguish the contributions of Cl ligands and PPh₃ ligands to HOR activity. In the Pt core of Pt₆NCs, there are three types of Pt atoms with different coordination environments, namely (1) Pt-P, (2) Pt-Cl, and (3) P-Pt-Cl (Supplementary Figure 27). It is found that the ΔG_{*H} of site 1 (-0.07 eV) for Pt₆NCs is very close to the ideal value ($\Delta G_{*H} = 0$) for HOR. In comparison, the ΔG_{*H} values of site 2 (+0.24 eV) and site 3 (+0.46 eV) reveal larger deviation from the ideal value. These results indicate that the PPh₃ ligands can significantly optimize the HBE of Pt compared to the Cl ligands.”

3. The authors discussed *OH reaction with CO for the removal. Would *OH also remove PPh₃ and Cl on Pt₆ (hence, partially deprotected Pt₆ and better activity)? While the fully-deprotected Pt₆NCs/C-550 is demonstrated to be not good, what about a brief thermal treatment to partially deprotect Pt₆?

Reply: We thank the reviewer for the enlightening comments. In order to verify the possibility that the reviewer proposed, we performed both DFT and experiments. First, based on DFT simulation, the binding energies of Pt-OH, Pt-PPh₃, and Pt-Cl were calculated to be -2.81, -2.75, and -3.70 eV, respectively, which means that the Pt-OH binding strength is close to that of Pt-PPh₃ but weaker than that of Pt-Cl. This result suggests that there is a possibility that *OH species can compete with PPh₃ ligands. Moreover, XPS was further performed to examine the chemical states of Pt₆NCs/C catalyst before and after ADT. The XPS result reveals that the P 2p peak of Pt₆NCs/C after the HOR test is highly close to that of the initial sample (**Figure R13**), proving the stable molecular structure of the Pt₆NCs/C catalyst during the HOR process, thus ruling out the above possibility to a certain extent.

Figure R13. High-resolution XPS P 2p spectra of Pt₆NCs/C before and after ADT.

In addition, we also wondered whether partial deprotection of Pt₆NCs would result in higher HOR catalytic activity or not. To check this assumption, the Pt₆NCs/C was heat-treated at 350 °C (based on the TGA, **Figure R14**) for 5 h under Ar atmosphere, and the obtained sample was named Pt₆NCs/C-350. Notably, the binding energies of Pt 4f in Pt₆NCs/C-350 are between those of Pt₆NCs/C and Pt₆NCs/C-550 (**Figure R15**), indicating a weakening of the ligand effect and partial removal of the PPh₃ ligands. On the other hand, the P 2p XPS peak of Pt₆NCs/C-350 is more easily

detected than that of Pt₆NCs/C-550 and its binding energy is higher than that of Pt₆NCs/C. Based on XPS data, the Pt/C atomic ratio of Pt₆NCs/C-350 was calculated to be 2.73, higher than 1.45 for Pt₆NCs/C, demonstrating partial removal of P-containing ligands. These data validate that partial deprotection of Pt₆NCs was successfully achieved.

Figure R14. TGA of the PPh₃ under Ar atmosphere.

Figure R15. (a) High-resolution XPS Pt 4f spectra of Pt₆NCs/C, Pt₆NCs/C-350 and Pt₆NCs/C-550. (b) High-resolution XPS P 2p spectra of Pt₆NCs/C, Pt₆NCs/C-350 and Pt₆NCs/C-550.

Subsequently, the HOR activity of Pt₆NCs/C-350 catalyst was evaluated in H₂-saturated 0.1 M KOH aqueous solution. As shown in **Figure R16a**, the anodic current response speed of Pt₆NCs/C-350 is inferior to that of Pt₆NCs/C in the kinetic and diffusion-controlled regions. The half-wave potential for Pt₆NCs/C-350 catalyst at 2500 rpm reaches 27.5 mV, which is worse than that of Pt₆NCs/C (11.5 mV), suggesting a decrease in catalytic activity. HOR polarization curves of Pt₆NCs/C-350 at various rotating rates from 400 to 3600 rpm were recorded in **Figure R16b**. The geometric j_k of 5.2 mA cm⁻² at the overpotential of 50 mV was obtained for Pt₆NCs/C-350 catalyst according to the Koutecky-Levich equation (**Figure R16c**), which is much lower than that of 18.2 mA cm⁻² for Pt₆NCs/C. The geometric j_0 is also determined from the linear fitting of the micro-polarization region (**Figure R16d**). The Pt₆NCs/C-350 catalyst delivers a geometric j_0 of 2.39 mA cm⁻², which is far lower than 3.23 mA cm⁻² for Pt₆NCs/C. In conclusion, the HOR activity of the partially deprotected Pt₆NCs/C-350 is far less than that of our reported Pt₆NCs/C, confirming that ligands are indispensable for contributing high HOR catalytic activity of Pt₆NCs/C.

Figure R16. (a) HOR polarization curves of Pt₆NCs/C-350 and Pt₆NCs/C catalysts in H₂-saturated 0.1 M KOH solutions with the rotation speed of 2500 rpm at a scan rate of 5 mV s⁻¹. (b) HOR polarization curves and various rotating speeds of Pt₆NCs/C-350 catalyst. (c) the Koutecky-Levich plot of Pt₆NCs/C-350 catalyst at an overpotential of 50 mV (vs. RHE). (d) Linear current potential region around the equilibrium potential of Pt₆NCs/C-350. The dotted lines indicate the linear fitting of the data.

4. On the CO poisoning, it is related to the question of Pt-PPh₃ binding strength. Typically, PPh₃ should be stronger than CO in binding, but the authors' results seem to indicate that PPh₃ is weaker than CO. Can the authors compare the binding energy of PPh₃ and CO on the Pt₆?

Reply: Thank you so much for the valuable question. Indeed, CO containing lone pair electrons tends to form coordination compounds with transition metal atoms through coordination bonds. Therefore, it is necessary to discuss whether the introduction of CO can reduce the structural stability of Pt₆NCs. To compare the adsorption energies of CO, PPh₃, and Cl on the surface of Pt₆NCs, we constructed and optimized these adsorption models. The adsorption energies of CO, PPh₃, and Cl on the surface of Pt₆NCs increase gradually, which are -1.87, -2.75, and -3.70 eV, respectively, indicating that the adsorption of CO on the surface of Pt₆NCs is weaker than those of PPh₃ and Cl ligands. This result also means that the adsorbed CO molecules cannot affect the molecular properties and structural stability of Pt₆NCs during CO poisoning experiments.

5. In Suppl Fig 26, did the various sites of Pt₆ show similar adsorption energy for *H or is there a specific site for an optimum H-adsorption?

Reply: We thank the reviewer for the helpful question. In fact, different coordination environments are destined to form different *d*-orbital electron cloud distributions, thereby modulating the electronic coupling between the adsorbate and the catalyst. We divided the adsorption sites into (1) Pt-P, (2) Pt-Cl, and (3) P-Pt-Cl according to the different coordination structures (**Figure R10**). As shown in **Table R1**, such three coordination environments result in differential *d*-band density of states and intermediate adsorption energies. Among them, the Pt site in the Pt-P bonding possesses the best HBE and OHBE, which is most favorable for alkaline HOR catalysis. Therefore, we chose the Pt-P site as the main catalytic site for a detailed discussion in the manuscript.

In this revision, we have added these new data and corresponding discussion in the revised manuscript and Supplementary Information to clarify this aspect.

Revision:

Paragraph 1, Page 20

“In addition, it is found that the Gibbs free energy of *H (ΔG_{*H} for short) of Pt₆NCs (-0.07 eV; Fig. 4e) on the optimal adsorption site (Pt-P bonding) is very close to the ideal value ($\Delta G_{*H} = 0$) for HOR (Supplementary Fig. 27).”

Supplementary Figure 27, Page 17 in Supplementary Information

Supplementary Figure 27. Models of H adsorption on catalysts. (a) Pt₆NCs. (b) Pt₁SAs. (c) PtNPs. The gray, white, blue, pink, and red spheres represent C, H, Pt, P, and Cl atoms, respectively.

Supplementary Note IV, Supplementary Information

“**Supplementary Note IV:** Owing to the difficulties in distinguishing the effects of Cl ligands and PPh₃ ligands alone on HOR activity through an experimental approach, we resorted to DFT simulations to distinguish the contributions of Cl ligands and PPh₃ ligands to HOR activity. In the Pt core of Pt₆NCs, there are three types of Pt atoms with different coordination environments, namely (1) Pt-P, (2) Pt-Cl, and (3) P-Pt-Cl (Supplementary Figure 27). It is found that the ΔG_{*H} of site 1 (-0.07 eV) for Pt₆NCs is very close to the ideal value ($\Delta G_{*H} = 0$) for HOR. In comparison, the ΔG_{*H} values of site 2 (+0.24 eV) and site 3 (+0.46 eV) demonstrate larger deviation from the ideal value. These results indicate that the PPh₃ ligands can significantly optimize the HBE of Pt compared to the Cl ligands.”

Response to Reviewer #4's Comments:

Wang et al reported the synthesis, characterization, and HOR activity of Pt nanocluster decorated by PPh₃ ligand. In my view, the mechanism of HOR activity is well demonstrated with sufficient experimental data and calculation. I would like to recommend the publication of this article to Nature Communications after the authors address some minor issues:

Reply: We thank the reviewer for the positive assessment on our work and are grateful for his/her recommendation of publishing this manuscript in Nature Communications.

1. Is the coordination structure with the PPh₃ ligand still stable after HOR stability test? The MS spectra or other methods should be added for identification.

Reply: We appreciate the reviewer's valuable comment. In this work, Pt₆NCs were loaded on carbon

support in order to disperse Pt clusters, increase electrical conductivity, and reduce agglomeration during the HOR. In turn, Pt₆NCs are difficult to be separated from the carbon support surface, which makes the subsequent ESI-MS test difficult. Therefore, we turned to characterize the stability of Pt₆NCs by other methods.

After ADT, aged catalysts were carefully collected for XPS examination again. As shown in **Figure R17a**, the Pt 4*f* XPS spectrum of the aging Pt₆NCs/C manifests Pt 4*f*_{7/2} and Pt 4*f*_{5/2} at 72.2 eV and 75.4 eV, respectively. These peaks can be deconvoluted into two spin-orbit doublets, indicating the dominant population of Pt⁰ and the minor population of Pt²⁺, consistent with the initial Pt₆NCs/C catalyst. In addition, the peak of P 2*p* for aging Pt₆NCs/C is almost completely preserved, suggesting the stable existence of the PPh₃ ligand (**Figure R17b**). The size of Pt₆NCs/C after the HOR test was also characterized. HAADF-TEM images demonstrate that no marked size change is observed for the Pt₆NCs/C catalyst after long-term testing (**Figure R17c, d**). These results clearly indicate the structural robustness of our catalysts during the HOR test in alkaline electrolytes.

Figure R17. (a) High-resolution XPS Pt 4*f* spectra of Pt₆NCs/C before and after ADT. (b) High-resolution XPS P 2*p* spectra comparison of Pt₆NCs/C and after ADT. (c) TEM and (d) HAADF-STEM images of Pt₆NCs/C catalyst after ADT.

In this revision, **Figure R17** has been supplemented into the Supplementary Information as **Supplementary Figure 19**, and the corresponding discussion was also added to the manuscript.

Revision:

Paragraph 1, Page 16

“The surface chemistry and size change of Pt₆NCs/C after the ADT were also characterized (Supplementary Fig. 19). The similar Pt 4*f* and P 2*p* XPS spectra of Pt₆NCs before and after ADT corroborate the good structural stability of Pt₆NCs in the long-term HOR process (Supplementary Fig. 19a,b). HAADF-TEM images reveal that no marked size change could be observed for the Pt₆NCs/C catalyst after ADT (Supplementary Fig. 19c,d).”

Supplementary Figure 19, Page 13 in Supplementary Information

Supplementary Figure 19. (a) High-resolution XPS Pt 4f spectra of Pt₆NCs/C before and after ADT. (b) High-resolution XPS P 2p spectra comparison of Pt₆NCs/C before and after HOR test. (c) TEM and (d) HAADF-STEM images of Pt₆NCs/C catalyst after ADT.

2. Scale bar for charge density should be shown in Fig 6c. Two values only referring to red and blue colors cannot cover the complete information of this figure.

Reply: Thank you, and we have revised it accordingly in this revision.

Revision:

Figure 4c, Page 18

Figure 4c The slice perspective of differential charge density distribution for Pt₆NCs. In the electron density difference maps, the red and blue colors refer to the positive (0.05 e Å⁻³) and negative (-0.05 e Å⁻³) values, respectively.

Supplementary Figure 25 and 26, Page 16 in Supplementary Information

Supplementary Figure 25. (a) Charge distribution around the Pt₁SAs. Light yellow and blue areas denote charge density depletion and accumulation, respectively. (b) The slice perspective of differential charge density distribution for Pt₁SAs. In the electron density difference maps, the red and blue colors refer to the positive ($0.05 \text{ e } \text{Å}^{-3}$) and negative ($-0.05 \text{ e } \text{Å}^{-3}$) values, respectively. The gray, white, blue, and pink spheres represent C, H, Pt, and P atoms, respectively.

Supplementary Figure 26. (a) Charge distribution around the Pt₆NCs. Light yellow and blue areas denote charge density depletion and accumulation, respectively. (b-d) The slice perspective of differential charge density distribution for Pt₆NCs from different perspectives. In the electron density difference maps, the red and blue colors refer to the positive ($0.05 \text{ e } \text{Å}^{-3}$) and negative ($-0.05 \text{ e } \text{Å}^{-3}$) values, respectively. The gray, white, blue, pink, and reseda spheres represent C, H, Pt, P, and Cl atoms, respectively.

3. In Fig 4d, PDOS of *d* orbital should also be given.

Reply: Many thanks, and we have provided it in the revised manuscript according to the reviewer's good suggestion.

Revision:

Figure 4d, Page 18

Figure 4d The PDOSs of Pt-5d in Pt₁SAs, Pt₆NCs, PtNPs, and Pt₆NCs/C-550 (each *d*-band center is marked by a dashed line) with the Fermi level aligned at 0 eV; inset: computed corresponding catalyst models.

REVIEWERS' COMMENTS

Reviewer #1 (Remarks to the Author):

Based on the comments and suggestions of all reviewers, the authors carefully revised their manuscript, so I think this paper is acceptable.

Reviewer #2 (Remarks to the Author):

The revised paper by Wei Xing and coworkers is an improved version of their work giving answers to the comments. Hence, hereby I recommend the publication of the current version on Nature Communications.

Reviewer #3 (Remarks to the Author):

The authors performed additional DFT and experiments to address my questions. The R1 manu is now acceptable.

Reviewer #4 (Remarks to the Author):

The reviewers' comments are well addressed. I recommend the publication of this article on Nat Commun.

Response to Reviewers' Comments

Response to Reviewer #1's Comments:

Based on the comments and suggestions of all reviewers, the authors carefully revised their manuscript, so I think this paper is acceptable.

Reply: Thanks for the reviewer's time in reviewing our manuscript, and we are happy that the reviewer is satisfactory to our revision.

Response to Reviewer #2's Comments:

The revised paper by Wei Xing and coworkers is an improved version of their work giving answers to the comments. Hence, hereby I recommend the publication of the current version on Nature Communications.

Reply: Thanks for the comment. We are delighted to know that the reviewer is satisfied with the revision.

Response to Reviewer #3's Comments:

The authors performed additional DFT and experiments to address my questions. The R1 manu is now acceptable.

Reply: We appreciate the reviewer's positive comments as well as the approval for the acceptance of our manuscript.

Response to Reviewer #4's Comments:

The reviewers' comments are well addressed. I recommend the publication of this article on Nat Commun.

Reply: We much appreciate the reviewer's recognition of our work.